# Mitochondrial aconitase suppresses immunity by modulating oxaloacetate and the mitochondrial unfolded protein response

Eunah Kim[1,5], Andrea Annibal [2,5], Yujin Lee[1,5], Hae-Eun H. Park [1,5], Seokjin Ham[1,5], Dae-Eun Jeong[3], Younghun Kim[1], Sangsoon Park[1], Sujeong Kwon [1], Yoonji Jung[1], JiSoo Park[1], Sieun S. Kim[1], Adam Antebi [2,4] ✉ & Seung-Jae V. Lee [1] ✉

Accumulating evidence indicates that mitochondria play crucial roles in immunity. However, the role of the mitochondrial Krebs cycle in immunity remains largely unknown, in particular at the organism level. Here we show that mitochondrial aconitase, ACO-2, a Krebs cycle enzyme that catalyzes the conversion of citrate to isocitrate, inhibits immunity against pathogenic bacteria in *C. elegans*. We find that the genetic inhibition of *aco-2* decreases the level of oxaloacetate. This increases the mitochondrial unfolded protein response, subsequently upregulating the transcription factor ATFS-1, which contributes to enhanced immunity against pathogenic bacteria. We show that the genetic inhibition of mammalian *ACO2* increases immunity against pathogenic bacteria by modulating the mitochondrial unfolded protein response and oxaloacetate levels in cultured cells. Because mitochondrial aconitase is highly conserved across phyla, a therapeutic strategy targeting ACO2 may eventually help properly control immunity in humans.

Mitochondria are the primary organelles for energy production, and play central roles in biological processes, including metabolism, apoptosis, stress response, aging, and immunity[1]. Signals such as reactive oxygen species and mitochondrial DNA, which are generated as by-products of mitochondrial dysfunction, are transmitted to other organelles, including the cytosol and the nucleus, leading to the protection of cells from infection[2,3]. Studies have identified mitochondrial factors that regulate cellular immunity, but mitochondrial components that affect immunity at the organism level remain largely unknown.

The Krebs cycle, also called the tricarboxylic acid cycle or the citric acid cycle, is the major catabolic process that oxidizes fatty acids, amino acids, and pyruvate[4]. The Krebs cycle provides electrons in the form of NADH and $FADH_2$ and succinate to the electron transport chain (ETC) for generating ATP through oxidative phosphorylation. Krebs cycle intermediates are also crucial for the biosynthesis of various metabolites. Recent studies have shown that Krebs cycle intermediates play roles in key biological processes, including DNA methylation, post-translational modification, hypoxic responses, lipid metabolism, aging, and immunity[5–11]. However, the mechanisms by

[1]Department of Biological Sciences, Korea Advanced Institute of Science and Technology, Daejeon 34141, South Korea. [2]Max Planck Institute for Biology of Ageing, Joseph-Stelzmann-Strasse 9b, Cologne 50931, Germany. [3]Department of Life Sciences, Pohang University of Science and Technology, Pohang, Gyeongbuk 37673, South Korea. [4]Cologne Excellence Cluster on Cellular Stress Responses in Aging Associated Diseases (CECAD), University of Cologne, Cologne, Germany. [5]These authors contributed equally: Eunah Kim, Andrea Annibal, Yujin Lee, Hae-Eun H. Park, Seokjin Ham. ✉e-mail: aantebi@age.mpg.de; seungjaevlee@kaist.ac.kr

which the Krebs cycle metabolites regulate immunity remain poorly understood.

In the current study, we identified a Krebs cycle enzyme, mitochondrial aconitase (ACO-2), that converts citrate to isocitrate, with immunosuppressive properties. We found that the genetic or pharmacological inhibition of ACO-2 increased the survival of *Caenorhabditis elegans* infected with pathogenic bacteria by upregulating the mitochondrial unfolded protein response (UPR[mt]). We then showed that *aco-2* RNAi altered the levels of several Krebs cycle metabolites, including citrate, isocitrate, succinate, fumarate, and oxaloacetate. Among them, we demonstrated that decreased levels of oxaloacetate were responsible for the enhanced immunity caused by upregulation of the UPR[mt]. Furthermore, we showed that mammalian ACO2 inhibited immune responses via modulating the UPR[mt] and oxaloacetate levels following infection with pathogenic bacteria. Our data suggest that the evolutionarily conserved mitochondrial aconitase can act as a target for anti-bacterial therapy for improving human immunity.

## Results

### Inhibition of ACO-2/mitochondrial aconitase boosts *C. elegans* immunity against pathogenic bacteria

To identify the Krebs cycle enzymes that regulate bacterial immunity in *C. elegans*, we performed an RNAi screen targeting each of 10 genes encoding the Krebs cycle enzymes (Fig. 1a, b; Supplementary Fig. 1a–i). Among them, RNAi targeting aconitase 2 (*aco-2*), which encodes mitochondrial aconitase that catalyzes the conversion of citrate to isocitrate, greatly increased the survival of animals upon infection with *Pseudomonas aeruginosa* PA14, Gram-negative pathogenic bacteria (up to 102%, Fig. 1b, c, and 4i). In contrast, *citrate synthase 1* (*cts-1*) RNAi substantially reduced the survival of worms on PA14 (Fig. 1b; Supplementary Fig. 1a). RNAi knockdown of other tested Krebs cycle enzyme-encoding genes had small or no effects on susceptibility to PA14 (Fig. 1b; Supplementary Fig. 1b–i). These data suggest that different Krebs cycle enzymes distinctly affect immunity against PA14.

We focused our functional analysis on *aco-2*, the RNAi knockdown of which exerted the greatest effect on PA14 susceptibility (Fig. 1b, c; Supplementary Fig. 2a). As *aco-2* is an essential gene (National BioResource Project *C. elegans*, Japan), we wondered about the threshold to which *aco-2* was sensitive to manifesting a phenotype, by performing PA14 resistance assays using serially diluted *aco-2* RNAi (Supplementary Fig. 2b). We found that differently diluted *aco-2* RNAi (1, 1/4, and 1/32) affected the survival of animals on PA14 in a dose-dependent manner (Supplementary Fig. 2c). In contrast to *aco-2*, RNAi or a mutation targeting *aco-1*, which encodes a cytosolic aconitase[12], shortened or had small effects on the survival of worms upon PA14 infection (Fig. 1d; Supplementary Fig. 2d). Thus, the two aconitases appear to affect immunity differently. *aco-2* RNAi enhanced survival on PA14 independently of *cdc-25.1* RNAi (Supplementary Fig. 3a), which increases resistance against PA14 by reducing germline proliferation[13]. Thus, reduced fertility caused by *aco-2* RNAi (Supplementary Fig. 3b) does not seem to be responsible for the enhanced resistance to PA14. We also found that *aco-2* RNAi diminished the accumulation of GFP-labeled PA14 in the intestinal lumen without affecting PA14 intake (Fig. 1e–g; Supplementary Fig. 3c, d). In addition, *aco-2* RNAi increased the survival of animals on the big-lawn of PA14 (Fig. 1h, i), where the animals cannot avoid the pathogen[14], and under a fast-killing condition (Fig. 1j), elicited by toxins secreted from PA14[15]. *aco-2* RNAi-treated worms were also resistant to *Staphylococcus aureus*, Gram-positive pathogenic bacteria (Fig. 1k). Inhibition of ACO-2 by treatment with fluoroacetic acid, a pharmacological inhibitor of aconitase in the Krebs cycle[16], also increased the survival of animals on PA14 and *S. aureus* (Fig. 1l, m). Together, these data suggest that the inhibition of mitochondrial ACO-2 enhances immune responses to pathogenic bacteria in *C. elegans*.

### ACO-2 knockdown differentially regulates various pathogen response genes

We determined whether genetic inhibition of ACO-2 altered immune responses at the transcriptome level, by performing RNA sequencing (RNA seq) analysis. We compared the transcriptomes of control RNAi- and *aco-2* RNAi-treated animals fed with *E. coli* diets or PA14 (Fig. 2a). Principal component analysis of the transcriptome indicated the separation of samples based on the biological conditions (Fig. 2b). We then categorized the genes that were differentially expressed by PA14 infection and/or *aco-2* RNAi into four groups (Fig. 2c, d: Groups i to iv). Group i consisted of 798 genes upregulated by *aco-2* RNAi whose expression was increased by PA14 infection (Fig. 2c, d). Group ii included 297 genes upregulated by *aco-2* RNAi whose expression was decreased by PA14 infection (Fig. 2c, d). Group iii included 4 genes downregulated by *aco-2* RNAi, for which PA14 infection increased the expression, and Group iv contained 60 genes downregulated by *aco-2* RNAi whose expression was decreased by PA14 infection (Fig. 2c, d). We found that Groups i and ii were enriched in genes that were differentially expressed by PA14 infection, while Groups iii and iv being depleted (Fig. 2e). This result indicates that *aco-2* RNAi upregulates PA14-responsive genes that may contribute to resistance against the pathogen. We therefore further analyzed the Group i and ii genes. We showed that Group i genes were strongly associated with the term "Stress response", and in particular with "Pathogen" in the WormCat annotation[17] (Fig. 2f, g); this result was corroborated by upregulation of 319 genes associated with "Defense response to other organism (GO:0098542)" in the Gene Ontology (GO) database[18] (Supplementary Fig. 4a). In contrast, Group ii genes were not related to "Pathogen", but were strongly associated with "Detoxification" of "Stress response" (Fig. 2f, g), and "Collagen" of "Extracellular material" and overall "Metabolism" (Supplementary Fig. 4b). These data raise the possibility that *aco-2* RNAi upregulates Group i genes that are associated with pathogen responses to enhance immunity against PA14.

Because simply disrupting normal mitochondrial physiology may lead to protection against infection in general, we compared transcriptome changes caused by *aco-2* RNAi with those by other mitochondrial dysfunctions[19–24] (Fig. 2h; Supplementary Fig. 5). We found that upregulation of "Defense response to other organism" in the GO database caused by *aco-2* RNAi was the highest among the comparisons that we executed; the mean expression change of the genes resulting from *aco-2* RNAi was 3.2 fold, whereas those caused by other mitochondrial dysfunctions were at most 1.5 fold. In addition, by analyzing WormCat terms, we showed that genes upregulated by *aco-2* RNAi were highly enriched with the term "Pathogen" of "Stress response" (Fig. 2i; Supplementary Figs. 6, 7; Supplementary Data 4). In contrast, genes upregulated by mitochondrial protease-encoding *spg-7* RNAi were enriched with "Detoxification" of "Stress response", and those by *gas-1*, *clk-1*, or *nuo-6* mutations were related to "Collagen" of "Extracellular material" (Fig. 2i). Together, these data suggest that knockdown of *aco-2* results in a robust transcriptomic response to pathogen infection compared with other implementations that disrupt mitochondrial physiology.

Next, we analyzed the immune signaling pathways that participated in the upregulation of pathogen responses by *aco-2* RNAi, by comparing Group i genes with previously published data (Fig. 2j; Supplementary Figs. 8, 9). We found that Group i genes significantly overlapped with genes upregulated by the PMK-1 signaling axis, based on the enrichment of several key components in the pathway: PMK-1, SEK-1 (PMK-1-activating kinase), ATF-7 (transcription factor acting downstream of PMK-1), and VHP-1 (negative regulator of PMK-1) (Fig. 2j: marked with dots). The Group i genes also substantially overlapped with genes downregulated by NIPI-3 and ELT-2 (Fig. 2j). The Group i genes were also enriched in genes induced by *spg-7* depletion in an ATFS-1-dependent manner (Fig. 2j). Gene set enrichment analysis, which quantifies the overall expression changes of specific gene sets

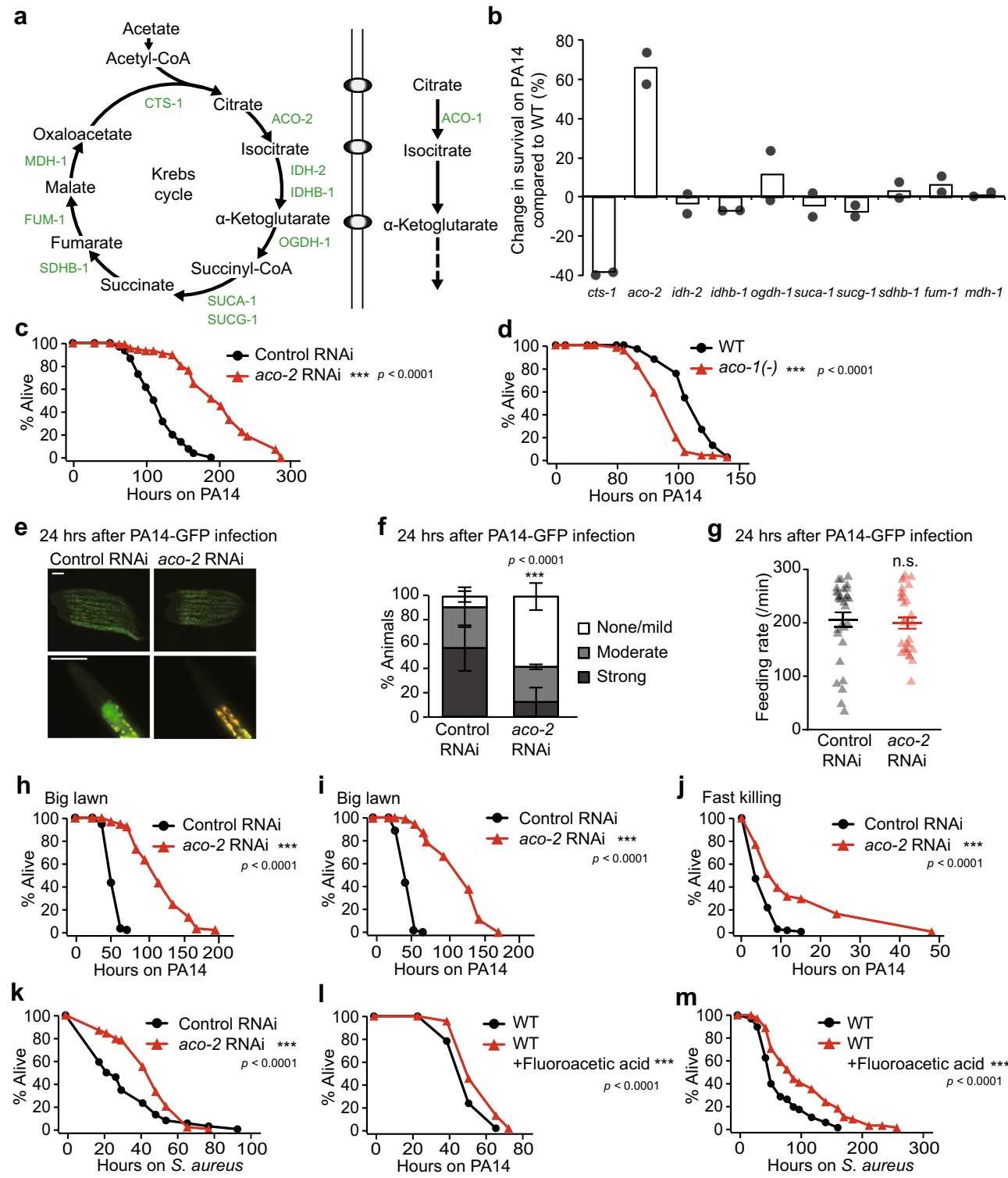

between two conditions[25], confirmed the strong upregulation of 9 out of 10 immune signaling gene sets by PA14 infection and *aco-2* RNAi (Supplementary Fig. 8b). Consistent with our transcriptome analysis, we found that the majority of selected targets of PMK-1/ATF-7 and ATFS-1 were upregulated by both PA14 infection and *aco-2* RNAi (Fig. 3a–l). In contrast, three selected targets of SKN-1/NRF, whose knockdown overlapped in both directions in our comparison with Group i genes (Fig. 2j), were upregulated by *aco-2* RNAi, but not by PA14 infection (Fig. 3m–o). These data suggest that *aco-2* RNAi upregulates immune responses via the PMK-1 signaling axis and/or ATFS-1.

### *aco-2* RNAi increases immunity via upregulating UPR[mt]

We investigated whether PMK-1/ATF-7 signaling or ATFS-1 contributed to the enhanced immunity caused by *aco-2* RNAi. We found that mutations in *atfs-1* fully suppressed the enhanced PA14 resistance of *aco-2* RNAi-treated animals (Fig. 4a). In contrast, mutations in *pmk-1* or *tir-1/SARM*, which encodes an upstream positive regulator of PMK-1[26], indiscriminately decreased the survival of *aco-2* RNAi- and control RNAi-treated animals on PA14 (Fig. 4b, c). In addition, we found that mutations in *atf-7* partially suppressed the resistance of *aco-2* RNAi-treated animals against PA14 (Fig. 4d). We also investigated other

**Fig. 1 | Inhibition of mitochondrial *aco-2* increases anti-bacterial immunity in *C. elegans*. a** Enzymes (green) and metabolites (black) in the Krebs cycle of *C. elegans*. **b** % mean survival changes by each RNAi upon PA14 infection (*N* = 180 per condition, two independent trials). **c, d** The survival of animals on PA14 was increased by *aco-2* RNAi (**c**) but decreased by *aco-1(jh131)* [*aco-1(-)*] mutations (**d**) (small-lawn slow-killing assays). WT: wild-type. *aco-1* RNAi and *aco-2* RNAi knocked down respective genes, without off-target and compensatory effects (Supplementary Fig. 2e). **e** Representative images of control RNAi- and *aco-2* RNAi-treated animals 24 h after PA14-GFP exposure. Yellow: autofluorescence. Scale bar: 100 μm. **f** Quantification of PA14-GFP levels in the intestinal lumen in panel (**e**) (*N* ≥ 30 per condition, three independent trials). Control RNAi-treated animals shown in panels (**e**) and (**f**) are the same experimental sets shown in Supplementary Fig. 3c, d. Error bars represent the SEM (***p < 0.001, two-sided Chi-squared test relative to control RNAi). **g** Feeding (pharyngeal pumping) rate of control RNAi- and *aco-2* RNAi-treated animals 24 h after PA14-GFP infection (*N* = 30 per condition, three

independent trials, two-tailed Student's *t*-test relative to control RNAi). **h–k** *aco-2* RNAi increased animal survival under big-lawn PA14-mediated slow-killing with (**h**) or without (**i**) 5-fluoro-2′-deoxyuridine (FUDR) treatment, and fast-killing (**j**) and *S. aureus* infection (**k**) conditions. **l, m** Supplementation with 0.1 mM fluoroacetic acid increased survival on the big-lawn of PA14 (**l**) and on *S. aureus* (**m**). *aco-2* RNAi-mediated immune responses were distinct from those of mitochondrial electron transport chain (ETC) components (see Supplementary Fig. 3h, i, Supplementary Data 1 for details). *aco-2* RNAi also increased resistance against oxidative and heat stresses (Supplementary Fig. 3j, k), with smaller effects than those on PA14 resistance. All the survival assays were performed at least twice independently. Asterisks in the survival curve panels indicate the significance of differences (***p < 0.001), calculated using a log-rank (Mantel-Cox method) test. See Supplementary Data 1 and 2 for additional repeats and statistical analysis for the survival and feeding assay data. Source data are provided as a Source Data file.

known transcription factors that regulate immunity, including SKN-1/NRF[27], DAF-16/FOXO[28,29], HLH-30/TFEB[30], ZIP-2/bZIP transcription factor[31], and HSF-1/heat shock factor 1[32], for effects on *aco-2* RNAi-mediated enhanced immunity. We found that mutations in *skn-1*, *daf-16*, *hlh-30*, *zip-2*, or *hsf-1* partially suppressed the increased survival of *aco-2* RNAi-treated animals upon PA14 infection, or had nonspecific effects (Fig. 4e–i). These data suggest that ATFS-1, a key transcriptional regulator of the UPR^mt and immunity[33–35] contributes to induced immune responses to PA14 by genetic inhibition of *aco-2*.

We further determined whether genetic inhibition of *aco-2* conferred a specific immunomodulatory role by comparing its effect with that of *spg-7* RNAi, which increases immunity[34]. We found that the effect of *aco-2* RNAi on the survival of animals on PA14 was greater than that of *spg-7* RNAi (Supplementary Fig. 10a). Interestingly, however, we showed that induction of *abf-2* and *hsp-6*, two selected UPR^mt targets, by *aco-2* RNAi was lower than that by *spg-7* RNAi (Supplementary Fig. 10b, c). Together with our RNA seq analysis (Fig. 2h, i), these data suggest that protection against pathogen infection by *aco-2* RNAi is a specific process, which does not seem to be caused by generally disrupting normal mitochondrial physiology.

### Decreases in oxaloacetate levels contribute to enhanced pathogen resistance by *aco-2* RNAi via UPR^mt signaling

We then asked which Krebs cycle metabolites were altered by the inhibition of ACO-2, contributing to the enhanced immunity. Using targeted metabolite analysis, we showed that *aco-2* RNAi increased the level of citrate, as reported previously[11], and reduced that of isocitrate, consistent with the role of ACO-2 in converting citrate to isocitrate[4] (Fig. 5a, b). *aco-2* RNAi also decreased the levels of succinate, fumarate, and oxaloacetate (Fig. 5a, b). We tested the effects of supplementation with each of the seven Krebs cycle metabolites—citrate, isocitrate, α-ketoglutarate, succinate, fumarate, malate, and oxaloacetate—on the survival of *aco-2* RNAi-treated animals on PA14 (Fig. 5c–i). We found that treatment with α-ketoglutarate or oxaloacetate consistently decreased the enhanced PA14 resistance caused by *aco-2* RNAi (Fig. 5e, i). In contrast, malate supplementation further increased the survival of *aco-2* RNAi-treated animals infected with PA14 (Fig. 5h). As *aco-2* RNAi significantly decreased the level of oxaloacetate, while increasing or marginally affecting that of α-ketoglutarate or malate (Fig. 5a, b), we further tested whether oxaloacetate affected the UPR^mt signaling, thus influencing immunity. We found that oxaloacetate decreased the induction of six selected ATFS-1 targets caused by *aco-2* RNAi upon PA14 infection (Fig. 6a–h). In contrast, α-ketoglutarate had small or no effects on the expression of the ATFS-1 targets (Supplementary Fig. 11a–h). Furthermore, oxaloacetate supplementation abrogated the induction of two tested UPR^mt targets, *abf-2* and *hsp-6*, in *aco-2* RNAi-treated animals, while partially suppressing that in *spg-7* RNAi-treated animals (Supplementary Fig. 10d, e). Thus, decreases in oxaloacetate levels appear to

specifically enhance immunity in *aco-2* RNAi-treated animals by upregulating the UPR^mt.

### Inhibition of mammalian ACO2 increases cellular immunity and UPR^mt

We then investigated the conservation of the role of mitochondrial aconitase in immunity in cultured mammalian cells. We asked whether mitochondrial aconitase ACO2 (Fig. 7a), the mammalian ortholog of ACO-2, affected immune responses during pathogen infection. We genetically inhibited mammalian *ACO2* using siRNA knockdown (*ACO2* KD) (Fig. 7b and Supplementary Fig. 12a), and found that *ACO2* KD increased the viability of HeLa cells upon infection with pathogenic *S. aureus* (Fig. 7c), consistent with the data using *C. elegans* (Fig. 1k, m). In contrast, knockdown of a cytochrome c oxidase subunit 5B (*COX5B*), the ortholog of *cco-1*, or citrate synthase 1 (*CS*), the ortholog of *cts-1*, did not (Fig. 7c and Supplementary Fig. 12b, c). These results suggest that *ACO2* KD conferred a specific immunomodulatory role, compared with other mitochondrial inhibition conditions. We further determined whether *ACO2* KD affected immune responses by measuring the expression of immune cytokines. We found that *ACO2* KD substantially increased the production of secreted IL6 and IL8, which was further elevated upon infection with *S. aureus* (Fig. 7d, e). *ACO2* KD also increased the mRNA levels of the immune cytokine genes, *IL6*, *IL8*, *IL1A*, *IL1B*, and *CXCL2*, which were further upregulated by *S. aureus* infection (Fig. 7f–j). In contrast, knocking down *CS* or *COX5B* did not cause a consistent effect on the expression of immune cytokine genes that we tested (Supplementary Fig. 12d–g). We then showed that enhanced immunity against *S. aureus* by siRNA knockdown of *Aco2* (*Aco2* KD) was recapitulated in mouse macrophage-derived RAW 264.7 cells (Supplementary Fig. 13a–f). Thus, reducing *ACO2* levels appears to elicit immune responses in cultured mammalian cells.

We tested whether *ACO2* KD increased immunity by upregulating the UPR^mt and modulating the level of oxaloacetate in mammalian cells, as we observed in *C. elegans*. We found that, upon infection with *S. aureus*, *ACO2* KD further increased the expression of *HSPD1*, *HSPA9*, and *CLIPP*, three of four selected targets of ATF5, a key mediator of the UPR^mt and the mammalian ortholog of *C. elegans* ATFS-1, in HeLa cells (Supplementary Fig. 12h–j). We further showed that supplementation with oxaloacetate suppressed the increased viability (Fig. 7k) and the induction of four out of five tested immune-response genes (Fig. 7l–p) conferred by *ACO2* KD upon *S. aureus* infection. We also confirmed these results with HeLa cells by using RAW 264.7 cells (Supplementary Fig. 13g–n). Taken together, these data suggest that the inhibition of mitochondrial aconitase increases immunity against pathogenic bacteria by regulating the UPR^mt and the levels of oxaloacetate in cultured mammalian cells as well as in *C. elegans*.

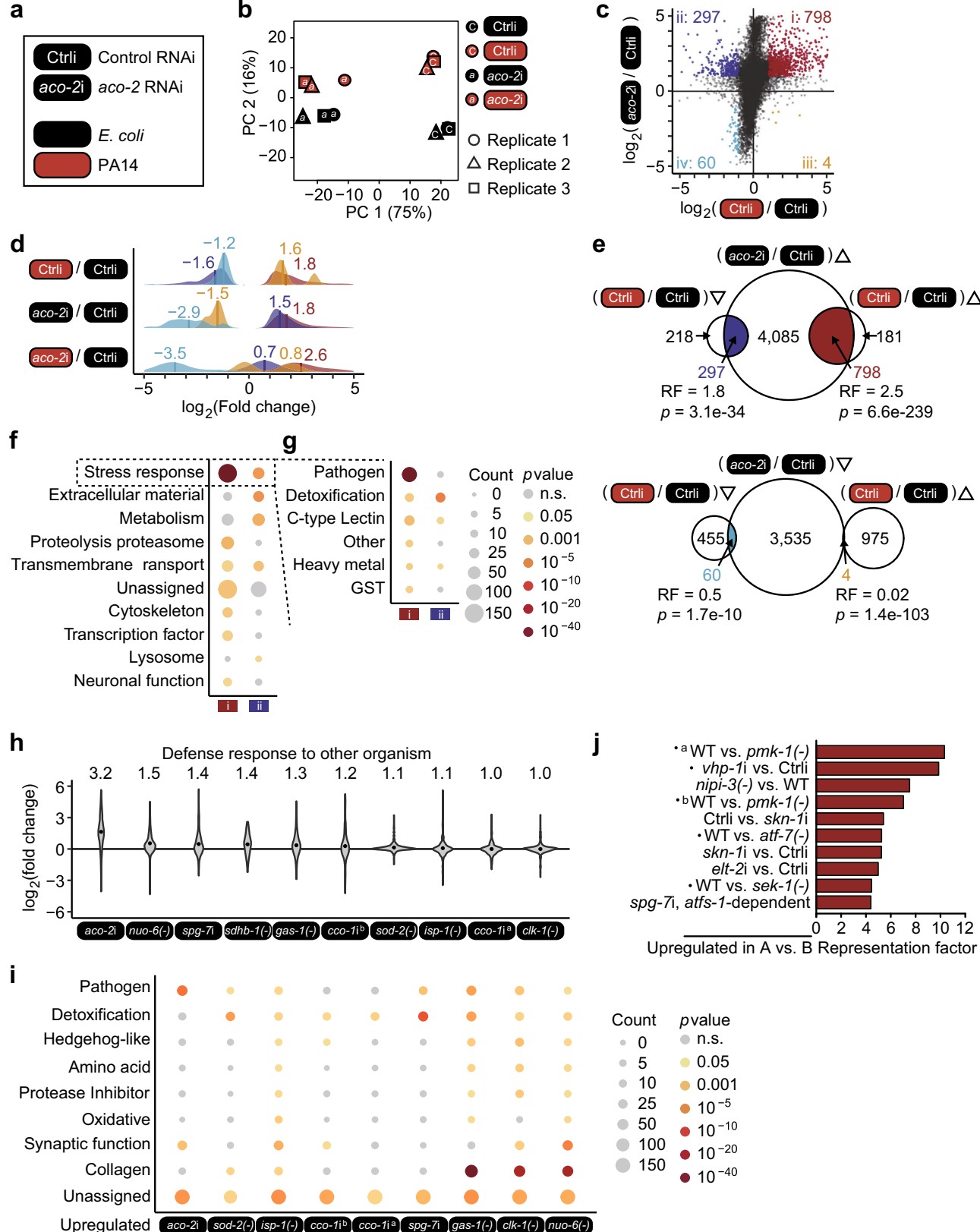

## Discussion

Mitochondria play crucial roles in multiple cellular processes that are essential for energy and metabolic homeostasis. Recent studies have shown that mitochondria are also critical for cellular immunity by influencing metabolic and physiological states[2,3,36]. However, in vivo research into the role of mitochondrial components and metabolites

in immunity at the organism level has been scarce. In this study, we identified mitochondrial aconitase as a negative regulator of immunity against pathogenic bacteria in *C. elegans*. We showed that the genetic inhibition of *aco-2* decreased the level of oxaloacetate, a Krebs cycle metabolite, subsequently enhancing the UPR[mt] by upregulating ATFS-1. Strikingly, we found that the increased immune response against

**Fig. 2 | Knockdown of *aco-2* upregulates various pathogen response genes.**
**a** Experimental variables used for our RNA-seq analyses [control RNAi (Ctrli) and *aco-2* RNAi (*aco-2*i), black: *E. coli*, red: PA14]. **b** A principal component (PC) analysis showing the separation of samples based on indicated biological conditions. **c** A scatter plot showing transcriptomic changes conferred by PA14 infection (*x* axis) and *aco-2* RNAi (*y* axis). PA14 infection respectively increased and decreased the expression of 798 genes (Group i) and 297 genes (Group ii) that were upregulated by *aco-2* RNAi, whereas it respectively increased and decreased the expression of 4 genes (Group iii) and 60 genes (Group iv) that were downregulated by *aco-2* RNAi. Absolute fold change > 2, two-sided test, Benjamini and Hochberg (BH)-adjusted *p* < 0.05. **d** A Gaussian kernel density plot showing transcriptomic changes elicited by PA14 infection or *aco-2* RNAi, categorized to Groups i, ii, iii, and iv. **e** Overlaps between genes that were differentially regulated by PA14 infection and genes that were upregulated by *aco-2* RNAi (Groups i and ii) (top) and genes that were downregulated by *aco-2* RNAi (Groups iii and iv) (bottom). RF: representation

factor. *p* values were calculated by using two-sided hypergeometric test. **f**, **g** Overrepresented WormCat terms at high levels (**f**), including "stress response" (**g**) of genes included in Group i and Group ii. *p* values were calculated by using two-sided hypergeometric test. **h** Comparison of expression changes caused by *aco-2* RNAi and those by other mitochondrial dysfunctions in genes associated with GO term "Defense response to other organism (GO:0098542)". Black dots represent average values. Average fold change is shown on top of each condition. Conditions were sorted based on average fold changes in a descending order. **i** Overrepresented WormCat terms at intermediate levels (category 2) of genes upregulated by *aco-2* RNAi and by other mitochondrial dysfunctions[19,21-24]. *p* values were calculated by using two-sided hypergeometric test. Ten representative terms were selected based on minimum *p* values among different conditions. **j** Representation factors of overlaps between Group i genes and immune signaling gene sets[§33,55,83-86] (See Supplementary Data 3 for details). §data (GSE82238) in GEO.

pathogenic bacteria by knockdown of mitochondrial aconitase was conserved in cultured mammalian cells, again in an oxaloacetate-dependent manner. These data indicate that mitochondrial aconitase, ACO-2/ACO2, which modulates the generation of Krebs cycle metabolites and the UPR[mt], can be targeted for the development of therapeutic strategies for achieving proper immunity.

Many previous studies using *C. elegans* have demonstrated that mitochondria are crucial for innate immunity against pathogenic bacteria[23,34,35,37-42]. The ATFS-1-mediated UPR[mt], which increases the production of mitochondrial chaperones, is crucial for immunity against infection by pathogenic bacteria[23,34,40,42]. As the majority of previous studies have focused on the relationship between the mitochondrial ETC and the UPR[mt] in immunity, the role of the Krebs cycle in immunity in *C. elegans* has remained unexplored. Thus, our current work showing increased immunity by inhibition of mitochondrial aconitase via ATFS-1 identifies a link between the Krebs cycle enzyme and the UPR[mt], which is important for organismal immunity. Further, while previous studies have focused mainly on the disbalance of mitochondrial protein subunits in triggering UPR[mt] and immunity[23,34,40,42], our work highlights the potential role of metabolic disbalance in immunity.

Accumulating evidence indicates that the Krebs cycle metabolites and their derivatives play key roles in many physiological processes, including immunity. For example, succinate, α-ketoglutarate, fumarate, itaconate, acetyl-CoA, and lactate affect immunity in mammalian cells by modulating transcription factors and epigenetic regulators, and by functioning as cytokine-like factors[8-10]. However, the role of oxaloacetate in immunity has remained obscure, despite reports showing physiological effects including the amelioration of hyperglycemia, liver injury and neuroinflammation in mammals, and the extension of lifespan in *C. elegans*[5,43-45]. Our data using genetic inhibition of *aco-2/ACO2* suggest that oxaloacetate can act as an immune modulator and can be used to mitigate inflammatory processes. Hypersensitivity against PA14 caused by *cts-1* RNAi, predicted to cause accumulation of oxaloacetate (Fig. 1a), is also consistent with the immunosuppressive role of oxaloacetate. Oxaloacetate may transmit mitochondrial signaling to suppress cellular and organismal resistance to pathogenic bacteria, because oxaloacetate participates in multiple metabolic processes, and consequently, cellular signaling pathways, in addition to its role in the Krebs cycle[4]. It is intriguing that oxaloacetate supplementation could ameliorate mitochondrial stress markers induced by *spg-7* RNAi (Supplementary Fig. 10d, e), revealing metabolic means to modulate such stress signaling pathways. Thus, our studies open up future work aimed at deciphering the metabolic and cellular processes through which oxaloacetate regulates immunity in *C. elegans* and mammalian cells.

In addition to its immune-regulating roles, α-ketoglutarate extends lifespan in *C. elegans* and mice[7,46]. Our data showing reduced PA14 resistance by α-ketoglutarate supplementation in *aco-2* RNAi-treated animals was unexpected (Fig. 5e), because of the strong

correlation between longevity and immunity[28,47-51]. One possible scenario is that α-ketoglutarate supplementation may increase the levels of other Krebs cycle metabolites, such as oxaloacetate, leading to the suppression of *aco-2* RNAi-mediated PA14 resistance. Another possibility is that α-ketoglutarate may directly reduce immunity in these animals, as α-ketoglutarate supplementation reduces inflammatory responses[46,52]. Inflammation, an essential immune response[53], occurs at the expense of normal physiological functions, and α-ketoglutarate may therefore decrease immunity in animals with reduced mitochondrial aconitase, while extending lifespan. Notably, several recent studies have uncovered potential tradeoffs between immunity and longevity[54-57]. Consistently, oxaloacetate, which we showed to suppress the *aco-2* RNAi-mediated enhanced immunity, increases lifespan in *C. elegans*[5]. In any case, this observation could indicate that specific metabolites within the Krebs cycle elicit specific immunomodulatory effects. Future research aimed at dissecting these possibilities will be essential for improving our understanding of the role of the Krebs cycle metabolites in immunity and longevity.

One limitation of our study using *C. elegans* is that Krebs cycle metabolites may indirectly influence pathogen responses. Our experimental scheme was designed to minimize the impact of the Krebs cycle supplements on the physiology of pathogenic bacteria PA14. Specifically, we performed fast-killing assays, in which the animals are rapidly killed, instead of slow-killing assays, to retain the impact of Krebs cycle metabolites that were provided before the survival assays. Importantly, at least for our experiments with cultured mammalian cells, we ruled out the effect of the metabolite supplementation on bacterial growth, because we added oxaloacetate in the last step of culture, before performing assays with *S. aureus*. Nevertheless, it will be crucial to devise experiments to directly test the effects of Krebs cycle metabolites on the immunity of animals.

Mitochondrial aconitase is crucial for mitochondrial function and cellular homeostasis, and therefore defects in this enzyme generally have pathological effects. Mutations in *ACO2* cause several diseases in humans, including infantile cerebellar-retinal degeneration[58] and optic neuropathy[59]. Downregulation of ACO2 is associated with various biological defects, including mitochondrial DNA damage, and progression of Huntington's disease and cancer[60-62]. The activity of mitochondrial aconitase is reduced during aging in mice and flies, contributing to decreases in lifespan[63-65], although our data indicate that proper knockdown of *aco-2* can increase lifespan in *C. elegans* (Supplementary Fig. 3h). Treatment with fluoroacetic acid, an inhibitor of aconitase, is also poisonous to aerobic organisms, because it irreversibly halts the Krebs cycle[16]. Consistent with these findings, homozygous *aco-2* deletion mutations cause sterility, embryonic lethality, and larval arrest during development. However, our work using *aco-2/ACO2* RNAi knockdown raises the possibility that controlled inhibition of mitochondrial aconitase may be used to improve immunity against pathogens. The increased immunity by *aco-2* RNAi

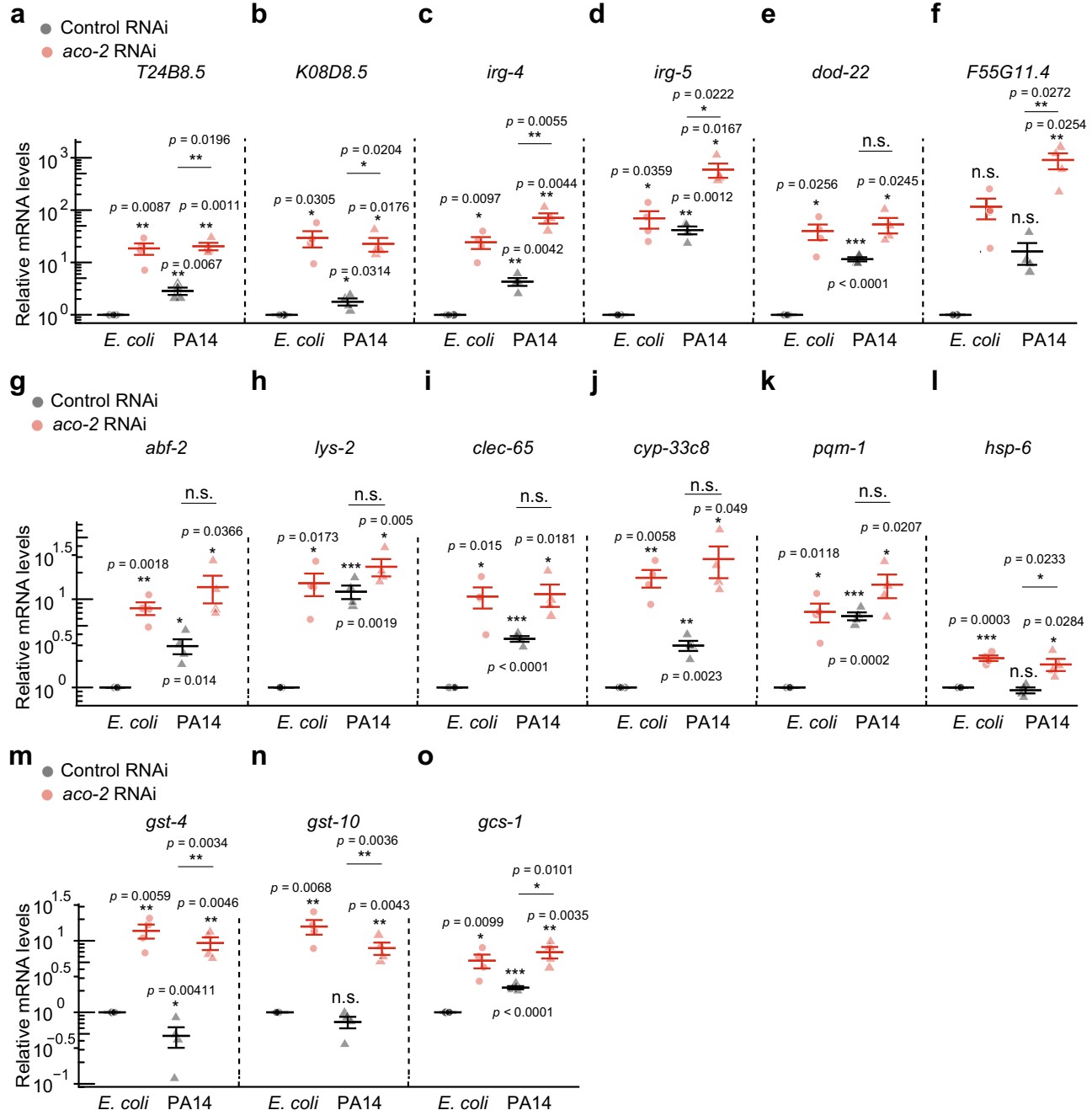

**Fig. 3 | *aco-2* RNAi and PA14 infection upregulates PMK-1/ATF-7 and ATFS-1 target genes. a–o** Relative mRNA levels of selected targets of PMK-1 and ATF-7, *T24B8.5* (**a**), *K08D8.5* (**b**), *irg-4* (**c**), *irg-5* (**d**), *dod-22* (**e**), and *F55G11.4* (**f**), ATFS-1, *abf-2* (**g**), *lys-2* (**h**), *clec-65* (**i**), *cyp-33c8* (**j**), *pqm-1* (**k**), and *hsp-6* (**l**), and SKN-1, *gst-4* (**m**), *gst-10* (**n**), and *gcs-1* (**o**) under indicated conditions using quantitative RT-PCR from four independent trials. Error bars indicate the standard error of the mean (SEM, \**p* < 0.05, \*\**p* < 0.01, \*\*\**p* < 0.001, n.s.: not significant, two-tailed Student's *t*-test relative to control RNAi under each bacterial condition). See Supplementary Data 5 for the details of primer sequences. Source data are provided as a Source Data file.

does not seem to be caused by a general reduction of activity of the Krebs cycle or mitochondrial function, based on multiple lines of evidence we obtained. First, different from *aco-2* RNAi, RNAi targeting any of the other components of the cycle did not produce substantial beneficial effects on immunity. Second, longevity-promoting knockdown of a mitochondrial ETC complex component, *cco-1*, did not elicit beneficial effects on immunity. Third, our RNA seq analysis indicates that *aco-2* RNAi causes a strong immune response at the transcriptomic level compared with other genetic impairments that disrupt mitochondrial physiology. Fourth, the impact of *aco-2* RNAi on the survival of animals on PA14 was greater than that of *spg-7* RNAi,

which efficiently induces UPR^mt. These data suggest that *aco-2* RNAi confers a specific immunomodulatory role distinct from perturbation of other mitochondrial components. It will be crucial to develop strategies for optimally and specifically inhibiting mitochondrial aconitase without causing detrimental impacts on the rest of the organism's physiological state.

## Methods

### *C. elegans* strains and maintenance

All the *C. elegans* strains were maintained at 20 °C on standard nematode growth medium (NGM) plates seeded with *E. coli* OP50

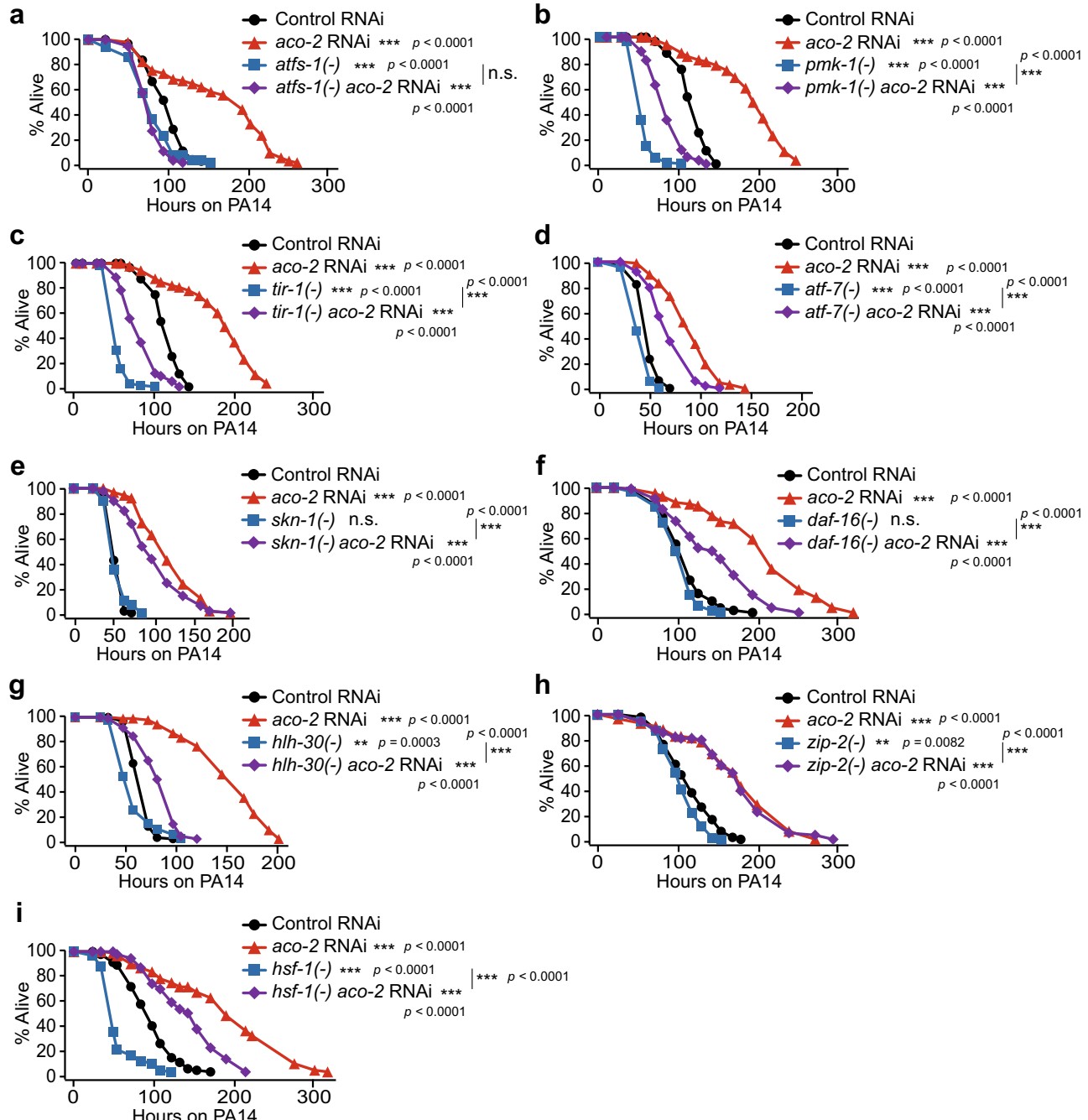

**Fig. 4 | *aco-2* RNAi increases immunity in an ATFS-1-dependent manner.**
**a**–**i** Shown are the survival curves determining the effects of *atfs-1(gk3094)* [*atfs-1(-)*] (**a**), *pmk-1(km25)* [*pmk-1(-)*] (**b**), *tir-1(tm3036)* [*tir-1(-)*] (**c**), *atf-7(qd22 qd130)* [*atf-7(-)*] (**d**), *skn-1(zj15)* [*skn-1(-)*] (**e**), *daf-16(mu86)* [*daf-16(-)*] (**f**), *hlh-30(tm1978)* [*hlh-30(-)*] (**g**), *zip-2(tm4067)* [*zip-2(-)*] (**h**), or *hsf-1(sy441)* [*hsf-1(-)*] (**i**) mutations on the survival of *aco-2* RNAi- and the control RNAi-treated worms against PA14 (slow-killing assay). Small-lawn PA14 killing assays were performed for panels (**a**), (**b**), (**c**), (**f**), (**h**), and (**i**), whereas big-lawn PA14 killing assays were performed for panels (**d**), (**e**), and

(**g**); the survival curves for control RNAi- and *aco-2* RNAi-treated animals are different for these panels because of the difference in the methods: small-lawn vs. big-lawn PA14 killing assays. All the survival assays were performed at least twice independently. Asterisks indicate the significance of differences (**$p < 0.01$, ***$p < 0.001$). n.s.: not significant. The $p$ values for survival data were calculated using a log-rank (Mantel-Cox method) test. See Supplementary Data 1 for additional repeats and statistical analysis for the survival assay data shown in this figure. Source data are provided as a Source Data file.

bacteria. Some strains were obtained from Caenorhabditis Genetics Center, which is funded by the NIH National Center for Resources (p40 OD010440), or National Bio-Resource Project (NBRP). Strains used in this study are as follows: N2 wild-type, IJ575 *atfs-1(gk3094) V* obtained by outcrossing VC3201 four times to Lee-laboratory N2, IJ130 *pmk-1(km25) IV* obtained by outcrossing KU25 four times to Lee-laboratory N2, IJ824 *tir-1(tm3036) III* obtained by outcrossing IG685 four times to Lee-laboratory N2, AU78 *agIs219[T24B8.5p::GFP::unc-54-3′UTR; ttx-*

*3p::GFP] III*, ZD318 *atf-7(qd22 qd130) agIs219[T24B8.5p::GFP::unc-54-3′ UTR; ttx-3p::GFP] III*, IJ1906 *hlh-30(tm1978) IV* obtained by outcrossing JIN1375 six times to Lee-laboratory N2, IJ1625 *skn-1(zj15) IV* obtained by outcrossing QV225 four times to Lee-laboratory N2, CF1042 *daf-16(mu86) I*, IJ134 *zip-2(tm4067) III* obtained by outcrossing FX4067 four times to Lee-laboratory N2, CF2495 *hsf-1(sy441) I*, SJ4100 *zcIs13[hsp-6::GFP]*, IJ975 *aco-1(jh131) X* obtained by outcrossing KJ550 four times to Lee-laboratory N2.

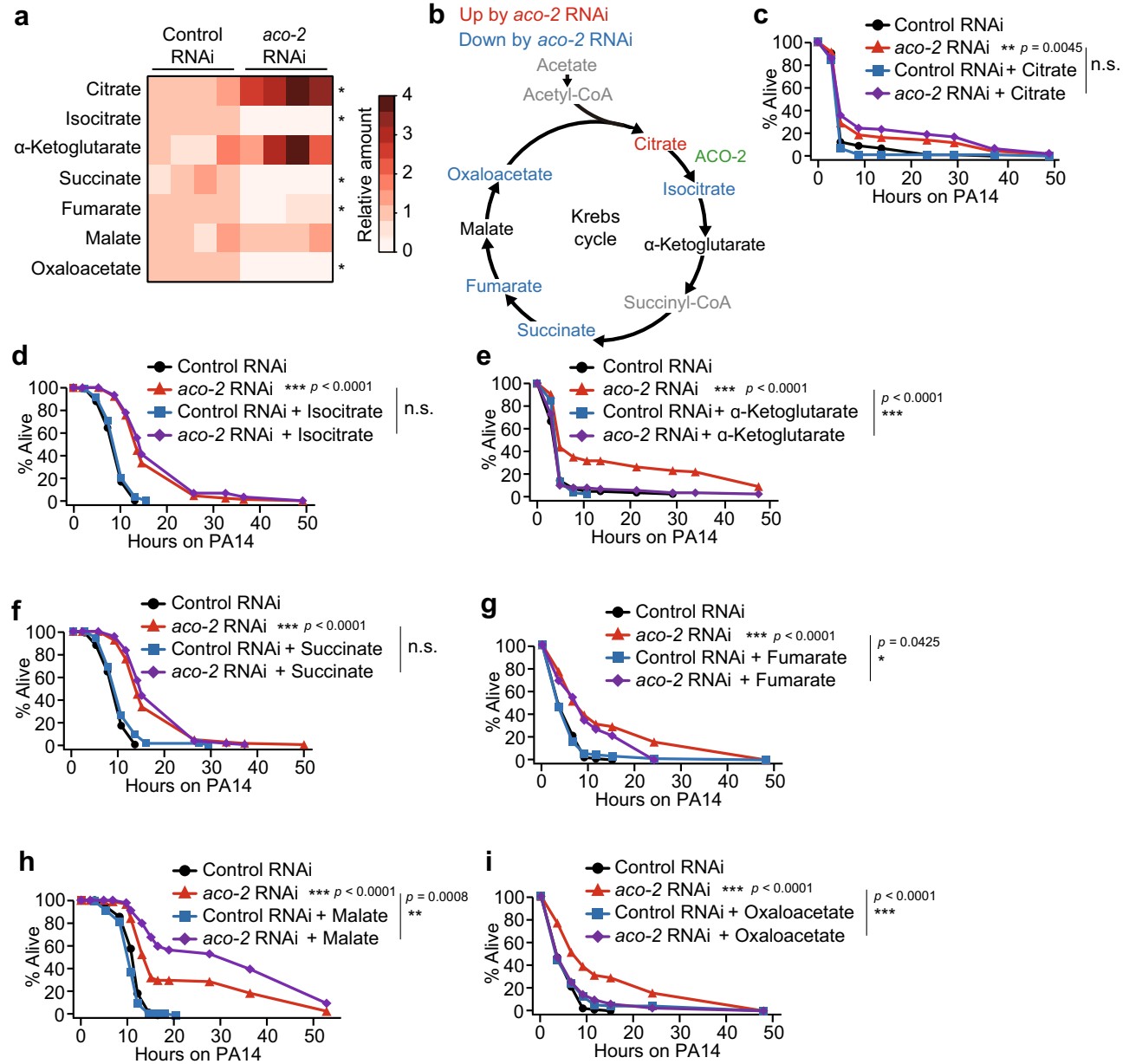

**Fig. 5 | Decreased oxaloacetate levels underlie enhanced pathogen resistance conferred by *aco-2* RNAi. a** Semi-quantitative analysis of Krebs cycle intermediates using liquid chromatography-mass spectrometry analysis from four independent trials (see Supplementary Data 6 and 7 for details of identified Krebs cycle metabolites). *$p < 0.05$, two-tailed Wilcoxon rank sum exact test. Citrate ($p = 0.03$), isocitrate ($p = 0.03$), α-ketoglutarate ($p = 0.06$), succinate ($p = 0.03$), fumarate ($p = 0.03$), malate ($p = 0.89$), and oxaloacetate ($p = 0.03$). **b** Summarized level changes of Krebs cycle intermediates by *aco-2* RNAi. Krebs cycle intermediates whose levels were significantly increased and decreased by *aco-2* RNAi were shown in red and blue, respectively. Black: no change. Gray: not determined. **c–i** The

survival of *aco-2* RNAi- and control RNAi-treated animals supplemented with each (8 mM) of citrate (**c**), isocitrate (**d**), α-ketoglutarate (**e**), succinate (**f**), fumarate (**g**), malate (**h**), and oxaloacetate (**i**) on PA14 (big-lawn fast-killing assay). All the survival assays were performed at least twice independently. Asterisks in survival curve panels indicate the significance of differences (*$p < 0.05$, **$p < 0.01$, ***$p < 0.001$). n.s.: not significant. The *p* values for survival data were calculated using a log-rank (Mantel-Cox method) test. See Supplementary Data 1 for additional repeats and statistical analysis for the survival assay data shown in this figure. Source data are provided as a Source Data file.

## RNAi induction

Double-stranded RNA-expressing HT115 bacteria were cultured in Luria broth (LB) containing 50 or 100 µg/ml ampicillin (USB, Santa Clara, CA, USA) overnight at 37 °C. One hundred microliter of RNAi bacterial culture was seeded onto NGM containing 50 or 100 µg/ml ampicillin, and incubated overnight at 37 °C. One mM isopropyl β-D-1-thiogalactopyranoside (IPTG, Gold biotechnology, St Louis, MO, USA) was added and incubated at room temperature for over 24 h before use. Julie Ahringer RNAi clones or Marc Vidal RNAi clones targeting *aco-2*,

*idh-2*, *idhb-1*, *ogdh-1*, *suca-1*, *sucg-1*, *sdhb-1*, and *mdh-1* were used[66,67]. RNAi constructs for targeting *cts-1* and *fum-1* were generated by infusion cloning (*cts-1*: 5′-attcgatatcaagctGTGTGTCAAGTCGCTCCACT and 5′-tatagggcgaattggTCAAGTGCTTCCATACCCAT/*fum-1*: 5′-attcgatatcaagct GTGAAGTCAACTCTCGTC and 5′-tatagggcgaattggAGCAGTGACAAG CATAAGTG). Cultures of HT115 containing empty vector (pL4440), *aco-2*, and *cdc-25.1* were prepared in LB broth containing 100 µg/ml ampicillin and grown overnight at 37 °C. Optical density at 590 nm values were adjusted to 0.9 and each bacteria culture mixed to 1:1 ratio.

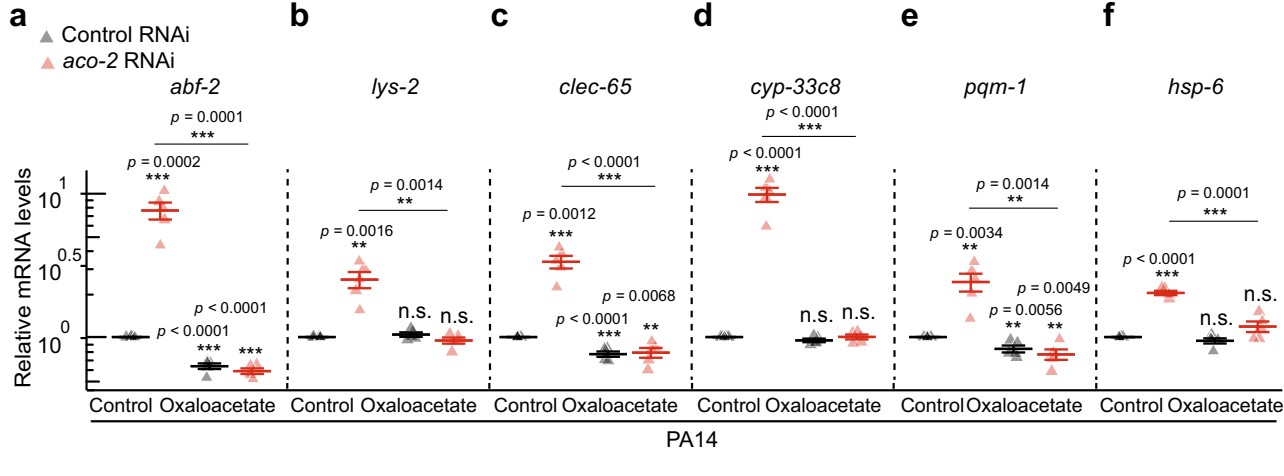

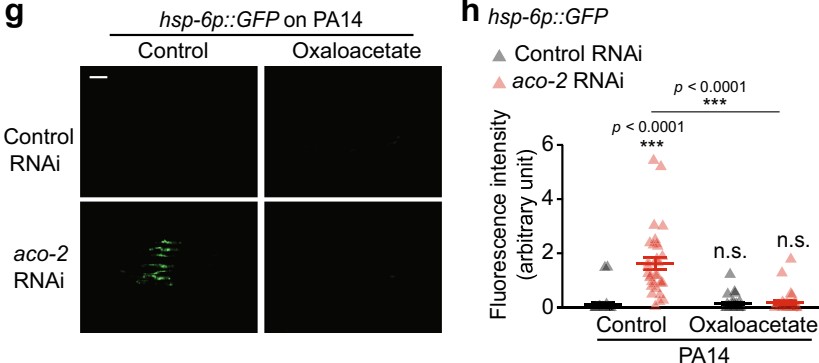

**Fig. 6 | Decreased oxaloacetate levels contribute to the induction of UPR^mt caused by *aco-2* RNAi. a–f** Oxaloacetate substantially decreased the induction of selected ATFS-1 targets, *abf-2* (**a**), *lys-2* (**b**), *clec-65* (**c**), *cyp-33c8* (**d**), *pqm-1* (**e**), and *hsp-6* (**f**), by *aco-2* RNAi upon PA14 infection by using quantitative RT-PCR from five independent trials. Oxaloacetate did not impair the knockdown efficiency of *aco-2* RNAi (See Supplementary Fig. 1i). *ama-1* and *tba-1* mRNA levels were used as normalization controls. **g** Representative fluorescence images of *hsp-6p::GFP* in control RNAi- and *aco-2* RNAi-treated animals exposed to PA14 without (Control) or with oxaloacetate treatment. Scale bar: 100 μm. **h** Quantification of fluorescence intensity of worms in panel (**g**) (N = 30 for control RNAi in control conditions, control RNAi and *aco-2* RNAi upon treatment with oxaloacetate, N = 31 for *aco-2* RNAi in control conditions, three independent trials). Control RNAi and *aco-2* RNAi data shown in panels (**g**) and (**h**) are the same experimental sets shown in Supplementary Fig. 11g, h. Error bars represent the SEM (\**p* < 0.05, \*\**p* < 0.01, \*\*\**p* < 0.001, n.s.: not significant, two-tailed Student's *t*-test relative to control RNAi-treated control conditions). See Supplementary Data 5 for the details of primer sequences. Source data are provided as a Source Data file.

## PA14 slow-killing assays

PA14 slow-killing assays were performed as previously described with minor modifications[68]. For PA14 small-lawn slow-killing assays, PA14 was cultured in LB media at 37 °C overnight, and 5 μl of the liquid culture was then seeded on the center of high-peptone NGM plates (0.35% bactopeptone). For PA14 big-lawn slow-killing assays, 15 μl of overnight-cultured liquid PA14 was seeded onto each high-peptone NGM plate and subsequently spread onto the entire surface of the plate. The PA14-seeded plates were incubated at 37 °C for 24 h and kept at 25 °C for 8 to 24 h before assays. L4 (>90%) and prefertile young adult (<10%) stage animals that were pre-treated with control or *aco-2* RNAi bacteria on NGM plates or with fluoroacetic acid[69] (Sigma, St. Louis, MO, USA) were transferred to the PA14-seeded plates with or without 50 μM 5-fluoro-2′-deoxyuridine (FUDR, Sigma, St. Louis, MO, USA), which prevents progeny from hatching; FUDR was used for standard slow killing assays in previous studies[68], but may affect proliferation and gut colonization of PA14. The animals were incubated at 25 °C, scored twice a day and counted as dead if the animals did not respond to prodding. Statistical analysis of survival data was performed by using OASIS (http://sbi.postech.ac.kr/oasis), OASIS2 (https://sbi.postech.ac.kr/oasis2) and GraphPad Prism (version 9, GraphPad Software, San Diego, CA, USA), which calculate *p* values using log-rank (Mantel-Cox method) test[70,71].

## PA14 fast-killing assays

PA14 fast-killing assays were performed as previously described with minor modifications[15,68]. Briefly, 15 μl of PA14 liquid culture was spread on peptone-glucose-sorbitol (PGS) plates without FUDR, incubated for 24 h at 37 °C, and kept at 25 °C for eight hrs before use. L4 stage larval wild-type animals that were pre-treated with control or *aco-2* RNAi on NGM containing each of the Krebs cycle intermediates (8 mM, pH 6.0, based on a previous study[7]) were transferred to PA14 on PGS plates for survival assays. NGM containing ddH₂O (pH 6.0) was used as a control for the Krebs cycle intermediate supplementation experiments.

## S. aureus killing assays

*Staphylococcus aureus* (MW2-WT) was used for infecting *C. elegans* as previously described with minor modifications[72]. Ten microliter of overnight-cultured bacteria were spread on the entire surface of the tryptic soy agar plates with 10 μg/ml nalidixic acid (Sigma, St. Louis, MO, USA). The plates seeded with *S. aureus* were incubated at 37 °C for 8 h and kept at 25 °C before assays. L4 stage animals that were pre-treated with control or *aco-2* RNAi bacteria on NGM plates or with fluoroacetic acid (Sigma, St. Louis, MO, USA) were transferred to the *S. aureus*-seeded plates with 50 μM FUDR. The animals were incubated at 25 °C, scored twice a day and counted as dead if the animals did not respond to prodding with a platinum wire.

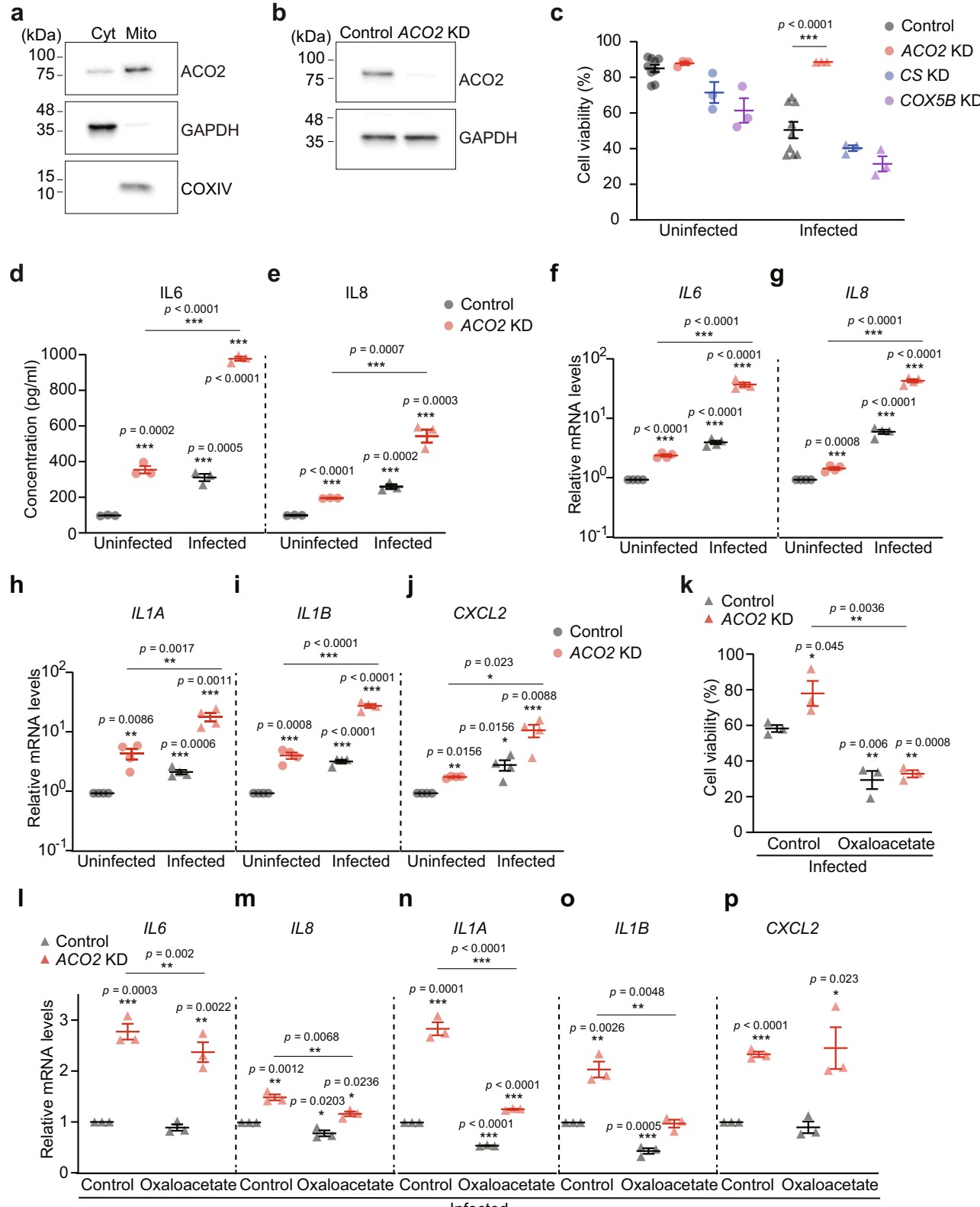

## Lifespan assays

Lifespan assays were performed as previously described[68]. Young adult animals that were cultured on control RNAi and *aco-2* RNAi plates from eggs were transferred to 5 μM FUDR-containing control and *aco-2* RNAi plates, respectively. The animals that did not move upon gentle touch with a platinum wire were counted as dead. The animals that crawled off the plates, displayed ruptured vulvae or internal hatching, or burrowed were censored but included in the subsequent statistical analysis.

## Stress resistance assays

Resistance assays against heat shock and oxidative stresses were performed as previously described with minor modifications[73]. Briefly, for oxidative stress assay, L4 stage animals were transferred onto 5 μM

**Fig. 7 | Downregulation of mammalian ACO2 increases cellular immunity by modulating the level of oxaloacetate. a** Western blots of fractionated lysates from HeLa cells (*n* = 3). GAPDH: cytosolic marker, COXIV: mitochondrial marker; cytosol (Cyt); mitochondria (Mito). **b** siRNA against *ACO2* (*ACO2* KD) efficiently decreased endogenous ACO2 protein levels in HeLa cells (*n* = 3). siRNA for non-target (Control) was used as a negative control. GAPDH was used as a loading control. **c** *ACO2* KD increased the viability of HeLa cells upon infection with *S. aureus* (*n* = 6), but *CS* KD (*n* = 3) or *COXSB* KD (*n* = 3) did not. **d, e** The amounts of IL6 (**d**) and IL8 (**e**) secreted from HeLa cells with *ACO2* KD or with control upon *S. aureus* infection were measured by using ELISA (*n* = 3). **f–j** Relative mRNA levels of

*IL6* (**f**), *IL8* (**g**), *IL1A* (**h**), *IL1B* (**i**), and *CXCL2* (**j**) upon treating with *S. aureus* in HeLa cells treated with Control or *ACO2* KD, measured by using quantitative RT-PCR (*n* = 4). **k** Treatment with 25 mM oxaloacetate suppressed the increased viability of HeLa cells by *ACO2* KD upon infection with *S. aureus* (*n* = 3). **l–p** Relative mRNA levels of *IL6* (**l**), *IL8* (**m**), *IL1A* (**n**), *IL1B* (**o**), and *CXCL2* (**p**) upon supplementation with oxaloacetate to *S. aureus*-infected HeLa cells treated with *ACO2* KD (*n* = 3). Error bars indicate the standard error of the mean (SEM, *$p < 0.05$, **$p < 0.01$, ***$p < 0.001$, two-tailed Student's *t*-test relative to control conditions). See Supplementary Data 5 for the details of primer sequences. Source data are provided as a Source Data file.

FUDR-treated NGM plates with *E. coli* bacteria and 7.5 mM t-BOOH (Sigma, St. Louis, MO, USA) solution. For the measurement of thermotolerance, L4 stage animals were placed in a 35 °C incubator. The number of live animals was counted as indicated, and recorded as dead when the animals did not respond to tactile stimuli with a platinum wire.

## RNA extraction and quantitative RT-PCR
RNA extraction and qRT-PCR were performed as previously described with minor modifications[68]. Wild-type animals fed with control RNAi and *aco-2* RNAi bacteria were cultured and harvested as described below in the "Generation of RNA seq data" section. Total RNA in *C. elegans* or cultured mammalian cells was isolated by using RNAiso plus (Takara, Seta, Kyoto, Japan), and was subsequently purified by using 75% ethanol. cDNA templates were synthesized by using ImProm-II™ Reverse Transcriptase kit (Promega, Madison, WI, USA) with random primers. Quantitative real-time PCR (qRT-PCR) was performed by using StepOne Real-Time PCR System (Applied Biosystems, Foster City, CA, USA) as described in the manufacturer's protocol. Comparative $C_T$ method was used for the quantitative analysis of mRNAs. For *C. elegans* data sets, the mRNA levels of *ama-1*, which encodes an RNA polymerase II large subunit, and *tba-1*, which encodes a tubulin α, were used as normalization controls. For the data sets using cultured mammalian cells, the mRNA level of human *GAPDH* or mouse *Gapdh* was used as a normalization control. The primer sequences used for qRT-PCR analyses in this study are listed in Supplementary Data 5.

## Generation of RNA seq data
Wild-type animals fed with control RNAi and *aco-2* RNAi bacteria were allowed to develop to L4 stage larvae at 20 °C and then transferred to control RNAi bacteria- or PA14-seeded big-lawn plates (high-peptone NGM plates with 0.35% bactopeptone) after washing with M9 buffer two to three times. After incubation at 25 °C for 12 h, animals were harvested by washing with M9 buffer two to three times to remove residual bacteria. Total RNA was extracted as described above in the "RNA extraction" section. The qualities of RNA samples were analyzed with 2100 Bioanalyzer (Agilent, CA, USA). The RNA integrity numbers, which indicate the quality of RNA samples, of all the samples were higher than > 7.0. Sequencing libraries were constructed by using TruSeq Stranded mRNA LT Sample Prep Kit (Illumina, CA, USA) and paired-end sequencing of Illumina NovaSeq 6000 was performed (Macrogen, Seoul, South Korea).

## Analysis of RNA seq data
Alignment and quantification of RNA seq data were performed based on the parameters described in the guidelines of ENCODE long RNA-Seq processing pipeline (https://www.encodeproject.org/pipelines/ENCPL002LPE/). Sequencing pairs were aligned to the *C. elegans* genome WBcel235 (ce11) and Ensembl transcriptome (release 103) by using STAR (v.2.7.0e)[74]. Genes with the aligned pairs were quantified by using RSEM (v.1.3.1)[75]. The raw counts were used for the principal component analysis and the identification of differentially expressed genes (DEGs). Genes with fold change > 2 and adjusted *p* value < 0.05 were identified as DEGs by using DESeq2 (v.1.22.2)[76]. Wald test *p* values

were adjusted for multiple testing using the procedure of Benjamini and Hochberg. Biological terms enriched in genes of interest were identified by using WormCat[17]. Expression changes of genes associated with the GO term "Defense response to other organism (GO:0098542)" were also determined[18]. To analyze through which immune signaling *aco-2* RNAi and PA14 infection mediated pathogen resistance, previously published gene sets were collected. Gene sets significantly enriched (*p* value < 10⁻³) in the GO term "Defense response to other organism" were subsequently chosen by using g:Profiler[77]. See Supplementary Data 3 for the details of the gene sets. Global expression changes of the immune signaling gene sets caused by *aco-2* RNAi and PA14 infection were represented as normalized enrichment scores (NES) by using gene set enrichment analysis (GSEA) (v.3.0)[25]. Gene sets whose false discovery rate *q* value < 0.05 in any comparison were regarded as significant in GSEA. Heatmaps were plotted to display gene expression changes caused by PA14 infection or by *aco-2* RNAi, and genes were clustered by hierarchical clustering. Jaccard similarity, which measures the ratio of intersection to union of gene sets, was calculated to examine distances among gene sets. Because mitochondrial dysfunctions generally slow development rates, which can be a confounding factor for the WormCat enrichment analysis, developmental time of each sample was estimated and adjusted to that of the control sample by using RAPToR [Real Age Prediction from Transcriptome staging on ref. 78]. Different thresholds for different datasets were chosen to obtain a comparable number of genes: fold change > 2 and *p* < 0.05 [863 genes by *aco-2* RNAi], (1) fold change > 2 and *p* < 0.05 [625 genes by *cco-1* RNAi], (2) fold change > 2 and *p* < 0.05 [644 genes by *cco-1* RNAi], (3) fold change > 2 and Benjamini and Hochberg (BH)-adjusted *p* < 0.05 [882 genes by *clk-1(qm30)*], fold change > 2 and BH-adjusted *p* < 0.01 [709 and 580 genes by *isp-1(qm150)* and *nuo-6(qm200)*, respectively], (4) fold change > 2 and *p* < 0.05 [829 genes by *gas-1(fc21)*], (5) fold change > 2 and *p* < 0.05 [394 genes by *spg-7* RNAi]. R (v.4.0.2, http://www.r-project.org) was used for all the data plotting and statistical tests unless stated otherwise.

## Semi-quantitative analysis of the Krebs cycle intermediates using LC-MS
The Krebs cycle intermediates were extracted as described previously[79]. Synchronized young adult animals were collected in four independent replicates and homogenized with a tissue lyser (Qiagen, Valencia, CA, USA) for 30 min at 4 °C. Protein concentrations were measured by using bicinchoninic acid (BCA) protein assay kit. The worm lysate was subjected to monophasic extraction by adding chloroform: methanol, 1:1 and incubated for 1 h at 4 °C, as previously described[80]. Samples were dried using a speed-vac and reconstituted in 90% aqueous acetonitrile. Six μl of samples were injected into a HILIC column (Zic-HILIC, 2.1 × 100 mm, 3.5 μm, Merck) coupled with Q exactive (Thermo Fischer Scientific GmbH, Bremen, Germany). Separation of the Krebs cycle analytes was obtained on a binary system following a modified gradient described in previous reports[81,82], using eluents A (acetonitrile/water, 95/5 v/v) and B (acetonitrile/water, 50/50, 10 mmol/L $KH_2PO_4$ pH 5.0). The duration of the gradient was 18 min, starting from 5 min with 35% eluent B and kept for 2 min, then ramped to 75% eluent B in 3 min and held for 2 min, subsequently in

3 min back to 0% eluent B and held for 3 min. The column was heated at 40 °C with a flow rate of 200 μl/min. Data were acquired in negative ion mode using a targeted-selected ion monitoring chromatogram mode (t-SIM). Using a resolution of 70,000 at 200 $m/z$, isolation window 1.0 Da, sheath and auxiliary gas values were set at 10 and 2 units respectively, voltage spray was set to 3.50 kV. Cysteamine S-phosphate sodium salt was used as an internal standard. Before each run the instrument was additionally calibrated using butylamine ($m/z = 72.08192$). Data were analyzed using Xcalibur 4.1 and quantified using Trace Finder 4.0.

## Microscopy

Pre-fertile young adult transgenic animals that expressed GFP were anesthetized by using 100 mM sodium azide (DAEJUNG, Siheung, Korea) or 4 mM levamisole (tetramisole; Sigma, St. Louis, MO, USA) and subsequently imaged by using AxioCam HRc (Zeiss Corporation, Jena, Germany) camera mounted on a Zeiss Axio Scope A.1 microscope (Zeiss Corporation, Jena, Germany). The fluorescence intensity of the animals was quantified by using ImageJ software (Rasband, W.S., ImageJ, U. S. National Institutes of Health, Bethesda, MD, USA, http://rsb.info.nih.gov/ij/). To determine the effect of *aco-2* RNAi on developmental time and brood size, synchronized wild-type eggs were allowed to grow on control or *aco-2* RNAi bacterial plates for 72 h. The images of the animals on the plates were captured by using a DIMIS-M camera (Siwon Optical Technology, Anyang, Korea).

## Intestinal PA14-GFP accumulation assays

Intestinal PA14-GFP accumulation assays were performed as previously described[40,73], with minor modifications. To measure the amount of PA14 that accumulated in the intestine, L4 animals that were pre-treated with RNAi were infected with PA14 that express GFP (PA14-GFP) for 24 or 36 h. "Big-lawn" PA14 that covered high-peptone NGM (slow-killing media) plates were used. Briefly, 15 μl of overnight cultured PA14-GFP in LB media that contain 50 μg/ml kanamycin (Sigma, St. Louis, MO, USA) was seeded and spread onto the NGM plates with 0.35% peptone. The plates were incubated at 37 °C for 24 h and subsequently at 25 °C for 24 h before use.

## Feeding rate measurements

Feeding (pharyngeal pumping) rates of animals were measured as described previously[73], with minor modifications. Ten animals cultured on each RNAi clone-expressing *E. coli* HT115 bacteria were transferred onto experimental plates containing PA14-GFP as indicated. After exposure with PA14-GFP for 24 or 36 h, the number of pumping for 30 s was counted by observing the pharyngeal pumping of an animal under a dissecting microscope, and the measurements were re-scaled to the number of pumping per min. Dead animals were excluded from the assays, and two-tailed Student's *t*-test was used for statistical analysis for the measurement of feeding at L4 stage.

## Mammalian cell culture and transfection

HeLa CCL-2 cells were obtained from American Type Culture Collection (ATCC). RAW 264.7 cells were a kind gift from Prof. Suk-Jo Kang (Korea Advanced Institute of Science and Technology, South Korea). These cells were cultured in Dulbecco's modified Eagle medium (DMEM) with high glucose (4 mM L-glutamine, 4.5 g/L glucose, sodium pyruvate; #SH30243.01, Cytiva, Marlborough, MA, USA) supplemented with 10% fetal bovine serum (#12483-020, Thermo Fisher Scientific, Waltham, MA, USA) and 100 U/ml of penicillin-streptomycin (#15140-122, Thermo Fisher Scientific, Waltham, MA, USA). Cells were maintained at 37 °C under 5% $CO_2$ and 85% humidity in a cell culture incubator. siRNA against *ACO2* (#L-009566-01, Horizon Discovery, Cambridge, UK), siRNA against mouse *Aco2* (#11429-1, Bioneer, Daejeon, South Korea), human *CS* (#1431-1, Bioneer, Daejeon, South Korea) and human *COX5B* (#1329-2, Bioneer, Daejeon, South Korea), and

nontargeting control siRNA (#D-001810-10, Horizon Discovery, Cambridge, UK), consisting of a pool of four designed control siRNAs, were used. HeLa cells ($1 \times 10^5$ cells) were seeded in six well plates 24 h before siRNA transfection. siRNA targeting *ACO2*, *CS*, or *COX5B*, or non-targeting control siRNA was transfected into HeLa cells using RNAi-MAX (#13778150, Thermo Fisher Scientific, Waltham, MA, USA) or Lipofectamine 3000 (#L3000015, Thermo Fisher Scientific, Waltham, MA, USA) transfection reagent. After incubation for 24 h, cells were infected with 10 multiplicity of infection (MOI) of *S. aureus* (MW2-WT) for 10 min at room temperature followed by 30 min at 37 °C with 5% $CO_2$ in a humid atmosphere. RAW 264.7 cells ($0.3 \times 10^5$ cells) were seeded in six-well plates 24 h prior to siRNA transfection. siRNA targeting *Aco2* or nontargeting control siRNA was transfected into RAW 264.7 cells using lipofectamine 3000 transfection reagent (#L3000001, Thermo Fisher Scientific, Waltham, MA, USA). After incubation for 24 h, cells were infected with 50 MOI of *S. aureus* (MW2-WT) for 10 min at room temperature followed by 30 min at 37 °C with 5% $CO_2$ in a humid atmosphere. After the medium was removed, the cells were incubated with DMEM with high glucose and gentamycin (10 μg/ml) for 24 h to remove extracellular bacteria, and the cells were harvested for the next experiments. *S. aureus* (MW2-WT) was cultured overnight at 37 °C in DMEM containing 2% FBS to reach a late logarithmic phase, and were diluted in DMEM with 10% FBS before supplementation to cells. To measure the MOI of *S. aureus*, infected cells were lysed with PBS containing 0.3% Triton X-100 at room temperature for 5 min, and then the diluted lysate was plated on tryptic soy agar (TSA) plates. The MOI of *S. aureus* was obtained by calculating colony forming units (CFU) per number of cells. Oxaloacetate (O7753, Sigma, St. Louis, MO, USA) was dissolved in phosphate-buffered saline (PBS) [137 mM NaCl, 2.7 mM KCl, 10 mM $Na_2HPO_4$, 1.8 mM $KH_2PO_4$, pH 7.4; #10010-049, Thermo Fisher Scientific, Waltham, MA, USA]. HeLa cells were treated with 25 mM oxaloacetate or PBS for 6 h before performing cell viability assays.

## Mitochondrial fractionation and western blot assays

Cells were washed with cold (4 °C) PBS twice and resuspended in 0.5 ml of fractionation buffer [20 mM HEPES (pH 7.4), 10 mM KCl, 2 mM $MgCl_2$, 1 mM EDTA, 1 mM EGTA] by scraping and incubated for 15 min on ice. The cell suspension was lysed by passing through 10 times using a 27-gauge needle and placed on ice for 20 min followed by centrifugation at $720 \times g$ for 5 min. The supernatant containing cytoplasm and mitochondria was re-centrifuged at $10,000 \times g$ for 5 min to separate mitochondria from cytoplasm. The mitochondrial pellet was lysed in Tris-buffered saline (TBS) [24.7 mM Tris-HCl (pH 7.5), 137 mM NaCl and 2.7 mM KCl] with 0.1% SDS by sonication with an amplitude of 20% using a 2 mm microtip for 3 s on ice. The samples were mixed with 4× laemmli sample buffer [62.5 mM Tris-HCl (pH 6.8), 10% glycerol, 1% LDS and 0.005% bromophenol blue] (#161-0747, Bio-rad, Contra Costa County, CA, USA) with 5% 2-mercaptoethanol (M3148, Sigma, St. Louis, MO, USA) and subsequently heated for 10 min at 95 °C and used for 10% SDS-PAGE. The proteins were then transferred to PVDF membranes (#10600021, GE healthcare, Chicago, IL, USA) at 300 mA for 1 h. The membranes were incubated with 5% bovine serum albumin solution in 1x TBS-T [TBS with 0.1% Tween 20] for blocking at room temperature for 30 min. The membranes were treated with primary antibodies against ACO2 (1:1000, ab129069, Abcam, Cambridge, UK), ACO1 (1:1000, ab183721, Abcam, Cambridge, UK), GAPDH (1:1000, G9545, Sigma, St. Louis, MO, USA), or COXIV (1:1000, 4850S, Cell Signaling Technology, Danvers, MA, USA) overnight at 4 °C. The membranes were then washed four times for 10 min using 1× TBS-T followed by incubating with secondary antibodies against rabbit antibodies (1:5000, #SA8002, ABfrontier, Seoul, South Korea) or mouse antibodies (1:5000, #SA8001, ABfrontier, Seoul, South Korea). After washing with TBS-T for 15 min, the membranes were then treated

with enhanced chemiluminescence (ECL) substrate (#1705061, Bio-Rad, Contra Costa County, CA, USA) for detecting protein bands. Images were visualized with ChemiDoc XRS+ system (Bio-Rad, Contra Costa County, CA, USA), and analyzed by using Image Lab software (Bio-Rad, Contra Costa County, CA, USA). The signal of ACO1, a control for cytoplasmic aconitase that exhibited a band size distinguishable from that of ACO2, remained upon membrane stripping for re-probing, when anti-ACO2 antibody was used in one out of three trials.

## Measurement of cell viability

Cell viability was measured by trypan blue exclusion assay. Cultured cells in six-well plates (#3335, Corning Costar, Corning, NY, USA) were washed with PBS to remove dead cells in suspension and treated with 300 µl of trypsin-EDTA (0.25%, #25200056, Thermo Fisher Scientific, Waltham, MA, USA) to detach cells from culture dishes. The cell suspension was then mixed with the same volume of 0.4% trypan blue solution (#15250061, Thermo Fisher Scientific, Waltham, MA, USA) and 10 µl of the samples were directly applied to counting slides (EVS-050, NanoEntek, Seoul, South Korea). Live and dead cells were counted by using an automatic cell counter (EVE, NanoEntek, Seoul, South Korea).

## Enzyme-linked immunosorbent assay (ELISA)

ELISA was performed for measuring the levels of IL6 and IL8 using human IL6 or IL8 uncoated ELISA kit (#88-8086 and 88-7066, Thermo Fisher Scientific, Waltham, MA, USA). Briefly, 96-well clear flat bottom microplates (#9018, Corning Costar, Corning, NY, USA) were incubated with 1:250 capture antibody diluted in 1× PBS overnight at 4 °C. The wells were washed three times with 200 µl of 1× PBS containing 0.05% Tween-20 and blocked with 200 µl of ELISA/ELISPOT Diluent (1×) at room temperature for 1 h. Recombinant human IL6 or IL8 proteins provided by the assay kits were reconstituted by the addition of triple distilled water and were serially diluted in ELISA/ELISPOT Diluent (1×), and used as a standard for quantitative ELISA. After discarding the blocking solution, the standard solution and the supernatants collected from cells were directly applied to ELISA reaction and incubated at room temperature for 2 h. The wells were then washed five times with 200 µl of 1× PBS containing 0.05% Tween-20, treated with detection antibody (1:250) diluted in ELISA/ELISPOT Diluent (1×), and incubated at room temperature for 1 h. After five times washing with 200 µl of 1× PBS containing 0.05% Tween-20, 1× HRP diluted in ELISA/ELISPOT Diluent (1×) was added to the wells and incubated at room temperature for 30 min. Subsequently, the wells were washed seven times and then 100 µl of 1× tetramethylbenzidine solution was added to the wells. After continuing the reaction for 15 min, 100 µl of 1 M $H_3PO_4$ was added to the wells. The absorbance was measured at 450 nm, and was subtracted with the value at 570 nm, a background signal, by using a microplate reader (SPARK, TECAN, Switzerland, Männedorf).

## Statistics and reproducibility

All the assays were conducted at least twice independently and statistical analysis used in this study is described in the figure legends and/or methods. No statistical method was used to predetermine the sample size. The samples were randomly allocated into experimental groups. All subcellular localization assays were performed double-blindly by at least two independent researchers. Other experiments, including survival assays in this study, were not performed blindly because of apparent phenotypic differences (e.g. different developmental time or movement) among conditions.

## Reporting summary

Further information on research design is available in the Nature Portfolio Reporting Summary linked to this article.

## Data availability

All raw and processed sequencing data generated in this study have been submitted to the NCBI Gene Expression Omnibus under accession number GSE181546. All relevant data, strains or plasmids are available from the corresponding author on reasonable request. Source data are provided with this paper.

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

## Acknowledgements

We thank *Caenorhabditis* Genetics Center (CGC) and National Bio-Resource Project (NBRP) for providing some *C. elegans* strains, Drs. Dennis H. Kim and Kwan Soo Ko for providing some pathogenic bacterial strains, and Dr. Suk-Jo Kang for providing RAW 264.7 cells. We thank Dr. Joo-Yeon Yoo and all the Lee lab members for comments on the manuscript and discussion. We thank Christian Latza for assisting A. Annibal with the metabolomics analysis. This work was supported by the National Research Foundation of Korea (NRF) grant funded by the Korea government (MSIT) NRF-2017R1A5A1015366 and NRF-2019R1A3B2067745 and a grant from the Korea Evaluation Institute of Industrial Technology (1415181231) to S.-J.V.L. and the Max Planck Society to A. Antebi.

## Author contributions

Conceptualization: E.K., A. Annibal, Y.L., H.-E.H.P., S.H., A. Antebi, and S.-J.V.L. Investigation: E.K., A. Annibal, Y.L., H.-E.H.P., S.H., D.-E.J., Y.K., S.P., S.K., Y.J., J.S.P., and S.S.K. Writing: E.K., A. Annibal, Y.L., H.-E.H.P., S.H., A. Antebi, and S.-J.V.L. Funding acquisition: A. Antebi and S.-J.V.L. Supervision: A. Antebi and S.-J.V.L.

## Competing interests

The authors declare no competing interest.
