## [Peer Review File · Nature Communications]

REVIEWER COMMENTS

Reviewer #1 (Remarks to the Author):

In the current manuscript, Kim et al show that knock-down or inhibition of the mitochondrial Krebs cycle aconitase (*aco-2*) enhances *C. elegans* immunity and increases resistance to the pathogenic *Pseudomonas aeruginosa* strain PA14. This is shown to be driven mainly by a decrease in oxaloacetate levels, and was further shown to be mediated by the UPRMT regulator ATFS-1. By comparing RNAseq data for worms exposed to *aco-2*-RNAi with datasets from other forms of mitochondrial disruption the authors conclude that *aco-2*(RNAi)-induced UPRMT and PA14 resistance represent an activation of a specific pathway which is distinct from the UPRMT induced by others forms of mitochondrial disruption.

The work described in this paper represents an advance in our knowledge of the outcomes of disrupting mitochondrial function, extending the known effects of electron transfer chain disruption on UPRMT induction to inhibition of the Krebs cycle. This is a significant advance, and the work described to support this is extensive and well performed. However, the conclusion that this inhibition activates a distinct regulatory program, could benefit from more support, by more detailed comparisons. This point, and additional comments are detailed below:

Major points:

- 1) The authors nicely consider the possibility that UPRMT induction following *aco-2* knock-down is a general induction of stress responses due to mitochondrial dysfunction, and continue to describe data that weakens this possibility, ending with the conclusion that *aco-2*(RNAi)-dependent responses are distinct (Fig. 2). They point out the higher fold induction in the *aco-2* response, as well as the difference in enriched annotations (lines 147-151), but a more detailed comparison is necessary to support this claim – which induced genes are shared among the responses, and which ones are distinct? This could be demonstrated using data that is currently included in Table S3, but this table should be arranged in a way more convenient for such comparisons, including the replacement of wormbase gene codes with common gene names. The authors should consider also alternative explanations for the differences between the different responses, such as the extent of mitochondrial disruption.
- 2) Related to point 1, the authors should also consider the possibility (or rule it out) that the induction of immune responses following *aco-2* KD is similar to activation of surveillance mechanisms described following disruption of other essential processes such as protein biosynthesis, which also include immune responses.

3) The authors highlight the response to *aco-2* KD as an immune response. Is lifespan without a pathogen similarly prolonged? Could it be that increased pathogen resistance is due to increased tolerance attributed to more general stress resistance?

4) The authors describe the contribution of tissue-specific *aco-2* KD to infection resistance as synergistic, but this is not clear from extended data Fig. 3, where differences between control and *aco-2* RNAi groups look marginal. The significance of differences is said to be reported in Table S1, but I couldn't find it there. I feel that this point is not essential for the conclusions of the paper, and is better off taken out. In general, it would be useful to have p-values included in the main text rather than only in a supplementary table.

5) The authors describe effects of *atfs-1* disruption as fully suppressing *aco-2*(RNAi)-dependent enhanced PA14 resistance (Fig. 4a), but it should be further pointed out that *atfs-1* disruption did not reduce resistance in control-treated animals. This is an important point to support the involvement of *atfs-1* only in the induced response.

6) Fig. 5a should show metabolite levels rather than z-scores. With that in mind, does oxaloacetate supplementation have any effect on the normal lifespan of *C. elegans* ?

Minor points.

1) To better represent the results shown for *aco-1* disruption, please rephrase the sentence “..the role of aconitase in immunity is therefore specific to the mitochondrial ACO-2.” to consider that the two are involved differently in immunity, as *aco-1* RNAi also affected pathogen resistance.

2) Lines 36-37 should be rephrased to more accurately describe the production of mitochondrially-generated signals as by-product of its function providing proxies for cellular homeostasis, rather the impression currently generated that these signals are produced to protect cells.

3) In Figure 1, when legends specify WT, does that mean worms treated with control RNAi? (which is the appropriate control). Please clarify. Also, in worms pre-treated with RNAi it is mentioned that 50 μ M FuDR were added to PA14 plates, but wouldn't that affect PA14 proliferation and gut colonization? Could the authors elaborate on that?

4) In extended data figure 4C, what is the distinction between the two PMK-1 regulated clusters? Also, in panel D, I can't find the circles representing the q-values. Also, if I understand correctly, here, as well as in Fig. 2, ELT-2 regulated genes examined are those that are UP-regulated after *elt-2* KD? i.e. ELT-2

repressed genes? and so are the NIPI-3 regulated genes. I would not define that as indicating anything about immune signaling. Why not focus on genes that are down-regulated following KD of these immune regulators?

Reviewer #2 (Remarks to the Author):

Kim et al. present a very interesting and timely manuscript that mitochondrial localized aconitase, by regulating the levels of oxaloacetate, can influence innate immunity. The manuscript is well written and the experiments are adequately defined and follow a logical progression that makes an excellent story. This manuscript is well-suited for Nature communications and will be of interest to the broad readership. In my opinion, the authors need only address one major concern and perhaps modify the text to address some minor issues. Overall, this is an very nice study that should be published following some suggested changes.

Major concerns:

the essential nature *aco-2* would suggest that even RNAi would make the animals sick and perhaps impact reproductive capacity. With this in mind the authors should investigate the developmental impact and reproductive consequences of *aco-2* RNAi as these are likely to impact the immune response documented. Developmental stage can greatly alter the response to pathogen and as such, animals with reduced *ACO-2* and slower development might not be stage appropriately for the assays. Similarly, reduced reproductive output has been shown by Fred Ausubel's group to greatly impact innate immune responses.

The authors should investigate how *aco-1* RNAi impacts *aco-2* and vice versa.

RNAi in neurons is very challenging, even in the enhanced neuronal RNAi strains. The authors should quantify RNAi impact in this tissue (perhaps with the GFP reporter).

Minor comments:

the authors should comment on the differential survival curves for WT and *aco-2* RNAi in figure 4. e.g., *aco-2* RNAi treated animals in Fig4d look like the WT in 4a and 4b for example.

The authors should take caution with epistatic relationship statements (same pathway, downstream, etc.). Most of the experiments in this manuscript are done with RNAi, which cannot be used for epistasis experiments.

REVIEWER COMMENTS

Reviewer #1 (Remarks to the Author):

In the current manuscript, Kim et al show that knock-down or inhibition of the mitochondrial Krebs cycle aconitase (*aco-2*) enhances *C. elegans* immunity and increases resistance to the pathogenic *Pseudomonas aeruginosa* strain PA14. This is shown to be driven mainly by a decrease in oxaloacetate levels, and was further shown to be mediated by the UPRMT regulator ATFS-1. By comparing RNAseq data for worms exposed to *aco-2*-RNAi with datasets from other forms of mitochondrial disruption the authors conclude that *aco-2*(RNAi)-induced UPRMT and PA14 resistance represent an activation of a specific pathway which is distinct from the UPRMT induced by others forms of mitochondrial disruption.

The work described in this paper represents an advance in our knowledge of the outcomes of disrupting mitochondrial function, extending the known effects of electron transfer chain disruption on UPRMT induction to inhibition of the Krebs cycle. This is a significant advance, and the work described to support this is extensive and well performed. However, the conclusion that this inhibition activates a distinct regulatory program, could benefit from more support, by more detailed comparisons. This point, and additional comments are detailed below:

Major points:

- 1) The authors nicely consider the possibility that UPRMT induction following *aco-2* knock-down is a general induction of stress responses due to mitochondrial dysfunction, and continue to describe data that weakens this possibility, ending with the conclusion that *aco-2*(RNAi)-dependent responses are distinct (Fig. 2). They point out the higher fold induction in the *aco-2* response, as well as the difference in

enriched annotations (lines 147-151), but a more detailed comparison is necessary to support this claim – which induced genes are shared among the responses, and which ones are distinct?

This could be demonstrated using data that is currently included in Table S3, but this table should be arranged in a way more convenient for such comparisons, including the replacement of wormbase gene codes with common gene names.

> We appreciate this reviewer's helpful comment. Following the suggestion by the reviewer, we compared genes upregulated by *aco-2* RNAi and those by other mitochondrial dysfunctions, among genes associated with WormCat term "Pathogen" of "Stress response". We added the comparison results to Supplementary Table S4 and visualized the results in Supplementary Figure 5b. We found that 24 out of 40 genes (60.0%) upregulated by *aco-2* RNAi were not upregulated by other mitochondrial dysfunctions. In contrast, only three genes (*T24B8.5*, *C49G7.5*, and *F22H10.2*) were commonly upregulated by four conditions, which are the maximum common conditions, including *aco-2* RNAi. These data are consistent with our functional data showing the specific role of ACO-2 in the pathogen responses. In addition, following the reviewer's the other suggestion, we included common gene names in another tab "Common_gene_names" of Supplementary Table S3 for better comparisons.

Results, page 7, line 137: "In addition, by analyzing WormCat terms, we showed that genes upregulated by *aco-2* RNAi were highly enriched with the term "Pathogen" of "Stress response" (Fig. 2i; Supplementary Fig. 5,6; Supplementary Table S4)."

Supplementary Table S4. The list of genes upregulated by mitochondrial dysfunctions and related to "Pathogen" of "Stress response" in WormCat.

Supplementary Figure 5, page 10, line 105: “**b**, Comparisons between genes upregulated by *aco-2* RNAi and other mitochondrial dysfunctions, among genes associated with WormCat term “Pathogen” of “Stress response”. We found that 24 out of 40 genes (60.0%) that were upregulated by *aco-2* RNAi were not upregulated by other mitochondrial dysfunctions. In contrast, only three genes (*T24B8.5*, *C49G7.5*, and *F22H10.2*) were upregulated by four conditions, which are the maximum common conditions, including *aco-2* RNAi. These data are consistent with our functional data showing the specific role of ACO-2 in the pathogen responses. See Supplementary Table S4 for the details of the gene sets.”

The authors should consider also alternative explanations for the differences between the different responses, such as the extent of mitochondrial disruption.

> We appreciate the reviewer’s comment. We sought to determine how the depletion of *aco-2* altered various other stress responses conferred by mitochondrial disruptions, in addition to “Defense response to other organism”. To this end, we analyzed multiple stress response pathways, effectively categorized in a recently published paper (Soo et al., 2022). Among them we chose five stress response pathways that were commonly upregulated by mitochondrial disruption conditions, including mutations in *clk-1*, *isp-1*, *nuo-6*, and *sod-2*: the mitochondrial unfolded protein response pathway (UPR^{mt}), the HIF-1-mediated hypoxia response pathway, the antioxidant gene expression, the DAF-16-mediated stress response pathway, and the p38-mediated innate immunity pathway. We found that upregulation of the three stress response pathways, the UPR^{mt}, the HIF-1-mediated hypoxia response pathway, and the antioxidant gene expression, by *aco-2* RNAi was not the highest

among the comparison. For example, our analysis indicated that *spg-7* RNAi conferred the highest upregulation of the UPR^{mt} among the comparisons; this is consistent with our qRT-PCR results of *abf-2* and *hsp-6* (Supplementary Figure 9b,c). In addition, *sdhb-1(R244H)* [*sdhb-1(-)*] caused the highest upregulation of the HIF-1-mediated hypoxia response pathway among the comparisons; this is consistent with the report showing the pronounced effects of the inhibition of SDH on HIF-1 α accumulation in the cytosol (Selak et al., 2005). We also found that *nuo-6* mutations [*nuo-6(-)*] elicited the highest induction of the antioxidant gene expression. We then showed that upregulation of the DAF-16-mediated stress response pathway and the p38-mediated innate immunity pathway caused by *aco-2* RNAi was the highest among the comparisons that we executed. Overall, these data provide an additional line of evidence for the specific effects of *aco-2* RNAi on pathogen responses. We added these points to our revised manuscript by revising the Results, and the Supplementary Figure 4 as follows.

Results, page 7, line 130: “Because simply disrupting normal mitochondrial physiology may lead to protection against infection in general, we compared transcriptome changes caused by *aco-2* RNAi with those by other mitochondrial dysfunctions^{19–24} (Fig. 2h; Supplementary Fig. 4).”

Supplementary Figure 4, page 7, line 75: “**Supplementary Figure 4. Stress response pathways affected by *aco-2* RNAi are different from those by general impairment of mitochondrial functions.**

a–e, Comparison of expression changes caused by *aco-2* RNAi and those by other mitochondrial dysfunctions for genes associated with multiple stress response pathways⁷: the mitochondrial unfolded protein response pathway (UPR^{mt}) (**a**), the HIF-1-mediated hypoxia response pathway (**b**), the

antioxidant gene expression (c), the DAF-16-mediated stress response pathway (d), and the p38-mediated innate immunity pathway (e). Black dots represent average values. Average fold change is shown on top of each condition. We found that upregulation of the three stress response pathways, the UPR^{mt}, the HIF-1-mediated hypoxia response pathway, and the antioxidant gene expression, by *aco-2* RNAi was not the highest among the comparison. For example, our analysis indicated that *spg-7* RNAi conferred the highest upregulation of the UPR^{mt} among the comparisons; this is consistent with our qRT-PCR results of *abf-2* and *hsp-6* (Supplementary Figure 9b,c). In addition, *sdhb-1(R244H)* [*sdhb-1(-)*] conferred the highest upregulation of the HIF-1-mediated hypoxia response pathway among the comparisons; this is consistent with the report showing the pronounced effects of the inhibition of SDH on HIF-1 α accumulation in the cytosol⁸. We also found that *nuo-6(qm200)* [*nuo-6(-)*] elicited the highest induction of the antioxidant gene expression. We then showed that upregulation of the DAF-16-mediated stress response pathway and the p38-mediated innate immunity pathway caused by *aco-2* RNAi was the highest among the comparisons that we executed. Overall, these data provide an additional line of evidence for the specific effects of *aco-2* RNAi on pathogen responses. See Legends of Fig 2, Supplementary Fig 8, and Supplementary Table S3 for specific references of data sets used in these analyses.”

2) Related to point 1, the authors should also consider the possibility (or rule it out) that the induction of immune responses following *aco-2* KD is similar to activation of

surveillance mechanisms described following disruption of other essential processes such as protein biosynthesis, which also include immune responses.

> We thank the reviewer for raising this important point. We examined whether the disruption of other essential processes (Melo and Ruvkun, 2012) altered “Defense response to other organism”. Specifically, we chose recently published datasets related to genetic inhibition of genes crucial for cell proliferation (*spo-11*, *htp-3*, and *tads-1*) (GSE199326; GSE110838), transcription (*hlh-2*) (GSE110835), RNA processing (*E02H1.1*, *prp-8*, *prp-19*, *skp-1*, and *snrp-200*) (GSE175363; Cartwright-Acar et al., 2022; GSE110836), and translation (*ife-2* and *ifg-1*) (Chomyshen et al., 2022; Soo et al., 2021). Among them, disruption of the majority of essential processes, including transcription, RNA processing, and translation, with the exception of cell proliferation (*spo-11*, *htp-3*, and *tads-1*), had small effects on the “Defense response to other organism”. These data provide another line of evidence for the specific effects of *aco-2* RNAi on pathogen responses, different from the possibility that impairment of essential processes can generally cause pathogen resistance. We added these points to our revised manuscript by revising the Results, and the Supplementary Figure 6 as follows.

Results, page 7, line 137: “In addition, by analyzing WormCat terms, we showed that genes upregulated by *aco-2* RNAi were highly enriched with the term “Pathogen” of “Stress response” (Fig. 2i; Supplementary Fig. 5,6; Supplementary Table S4).”

Supplementary Figure 6, page 12, line 116: “**Supplementary Figure 6. *aco-2* RNAi affects pathogen responses distinctly from the general impairment of essential processes.** Comparison of expression changes caused by *aco-2* RNAi and those by genetic inhibition of essential processes

for genes associated with GO term “Defense response to other organism (GO:0098542)”: cell proliferation (*spo-11*, *htp-3*, and *tads-1*) (GSE199326; GSE110838), transcription (*hlh-2*) (GSE110835), RNA processing (*E02H1.1*, *prp-8*, *prp-19*, *skp-1*, and *snrp-200*) (GSE175363; Ref¹⁰; GSE110836), and translation (*ife-2* and *ifg-1*)^{11,12}. Black dots represent average values. Average fold change is shown on top of each condition. Disruption of the majority of essential processes, including transcription, RNA processing, and translation, with the exception of cell proliferation, had small effects on the “Defense response to other organism”.

3) The authors highlight the response to *aco-2* KD as an immune response. Is lifespan without a pathogen similarly prolonged?

> We appreciate the reviewer’s critical comment. Related to this comment, in Supplementary Figure 2m, we showed that *aco-2* RNAi extended the lifespan of animals on a normal laboratory diet (*E. coli*) at 25°C (the same temperature for the pathogen survival assays). We found that the effect of *aco-2* RNAi on PA14 resistance (46%-102% increase: average 74%) is larger than that on lifespan (20%-31% increase: average 26%), as we described in the text and Supplementary Table S1.

Results, page 4, line 70: “Among them, RNAi targeting aconitase 2 (*aco-2*), which encodes mitochondrial aconitase that catalyzes the conversion of citrate to isocitrate, greatly increased the survival of animals upon infection with *Pseudomonas aeruginosa* PA14, Gram-negative pathogenic bacteria (up to 102%, Fig. 1b,c, and 4i).”

Discussion, page 17, line 346: "..., although our data indicate that proper knockdown of *aco-2* can increase lifespan in *C. elegans* (Supplementary Fig. 2m)."

Supplementary Figure 2, page 4, line 43: "m, *aco-2* RNAi extended the lifespan of worms at 25°C, the same temperature for pathogen survival assays; the effect of *aco-2* RNAi on PA14 resistance (46%-102% increase in 12 trials of PA14 small-lawn assays: average 74%) is larger than that on lifespan (20%-31% increase in two trials: average 26%) (Supplementary Table S1)."

Could it be that increased pathogen resistance is due to increased tolerance attributed to more general stress resistance?

> We appreciate the reviewer's comment. To determine whether *aco-2* RNAi knockdown increased general stress resistance as well as immunity in animals, we performed survival assays upon treating with two representative exogenous stresses, heat shock and oxidative stress. We found that *aco-2* RNAi increased resistance against heat and oxidative stresses. In the oxidative stress assay, *aco-2* RNAi increased resistance by 30%, 31%, and 45% in three trials. For the analysis of heat stress resistance at 35°C, *aco-2* RNAi increased the thermotolerance by 31% and 18% in two experiments. This is consistent with our RNA seq analysis, a positive correlation between stress resistance and immunity by *aco-2* RNAi. Thus, *aco-2* RNAi increases stress resistance and immunity, but the effect of *aco-2* RNAi on PA14 resistance (46%-102% increase in 12 trials of PA14 small-lawn assays: average 74%) is larger than that on stress resistance (30-45% increase in oxidative stress assays: average 38%, and 18-31% increase in thermotolerance: average

25%). We included these new data in our revised manuscript by revising the Figure 1 legends, and the Supplementary Figure 2 as follows.

Figure 1 legends, page 41, line 962: “*aco-2* RNAi also increased resistance against oxidative and heat stresses (Supplementary Fig. 2o–p), but the effects were smaller than those on PA14 resistance.”

Supplementary Figure 2, page 4, line 54: “**o**, *aco-2* RNAi increased the resistance of animals against oxidative stress [7.5 mM tert-butyl hydroperoxide (t-BOOH)]. **p**, *aco-2* RNAi increased the survival of animal under heat stress conditions (35°C).”

Methods, page 22, line 453: “**Stress resistance assays.** Resistance assays against heat shock and oxidative stresses were performed as previously described with minor modifications⁷³. Briefly, for oxidative stress assay, L4 stage animals were transferred onto 5 μM FUDR-treated NGM plates with *E. coli* bacteria and 7.5 mM t-BOOH (Sigma, St. Louis, MO, USA) solution. For the measurement of thermotolerance, L4 stage animals were placed in a 35°C incubator. The number of live animals was counted as indicated, and recorded as dead when the animals did not respond to tactile stimuli with a platinum wire.”

4) The authors describe the contribution of tissue-specific *aco-2* KD to infection resistance as synergistic, but this is not clear from extended data Figure 3, where differences between control and *aco-2* RNAi groups look marginal. The significance of differences is said to be reported in Table S1, but I couldn't find it there.

I feel that this point is not essential for the conclusions of the paper, and is better off taken out.

> We appreciate the reviewer's comment. We removed the tissue-specific RNAi results from the manuscript as the reviewer suggested.

In general, it would be useful to have p-values included in the main text rather than only in a supplementary table.

> We thank the reviewer for this comment. We added *p* values to all the figure panels with asterisks and described them in the figure legends.

5) The authors describe effects of *atfs-1* disruption as fully suppressing *aco-2*(RNAi)-dependent enhanced PA14 resistance (Fig. 4a), but it should be further pointed out that *atfs-1* disruption did not reduce resistance in control-treated animals. This is an important point to support the involvement of *atfs-1* only in the induced response.

> We appreciate the reviewer's comment. We now described the role of *atfs-1* in the induced immune response as follows.

Result, page 9, line 178: "These data suggest that ATFS-1, a key transcriptional regulator of the UPR^{mt} and immunity³³⁻³⁵, contributes to induced immune responses to PA14 by genetic inhibition of *aco-2*."

6) Fig. 5a should show metabolite levels rather than z-scores.

> We thank the reviewer for this comment. We have changed the data presentation of Figure 5A to show relative metabolite levels.

With that in mind, oxaloacetate supplementation have any effect on the normal lifespan of *C. elegans* ?

> A previous paper reported that oxaloacetate supplementation extends lifespan in *C. elegans* (Williams et al., 2009). Therefore, we discussed this issue in the revised manuscript as follows.

Discussion, page 15, line 318: “Consistently, oxaloacetate, which we showed to suppress the *aco-2* RNAi-mediated enhanced immunity, increases lifespan in *C. elegans*⁵”

Minor points.

1) To better represent the results shown for *aco-1* disruption, please rephrase the sentence “..the role of aconitase in immunity is therefore specific to the mitochondrial ACO-2.” to consider that the two are involved differently in immunity, as *aco-1* RNAi also affected pathogen resistance.

> We appreciate the reviewer’s comment. Following the suggestion by the reviewer, we rephrased the sentence as follows.

Results, page 5, line 86: “In contrast to *aco-2*, RNAi or a mutation targeting *aco-1*, which encodes a cytosolic aconitase¹², shortened or had small effects on the survival of worms upon PA14 infection (Fig. 1d; Supplementary Fig. 2d). Thus, the two aconitases appear to affect immunity differently.”

2) Lines 36-37 should be rephrased to more accurately describe the production of mitochondrially-generated signals as by-product of its function providing proxies for cellular homeostasis, rather the impression currently generated that these signals are produced to protect cells.

> Following the reviewer’s comment, we rephrased the sentence as follows.

Introduction, page 3, line 36: "Signals such as reactive oxygen species and mitochondrial DNA, which are generated as by-products of mitochondrial dysfunction, are transmitted to other organelles, including the cytosol and the nucleus, leading to the protection of cells from infection^{2,3}."

3) In Figure 1, when legends specify WT, does that mean worms treated with control RNAi? (which is the appropriate control). Please clarify.

> We thank the reviewer for this comment. We have changed WT to control RNAi in all the figures, tables and the text in our revised manuscript.

Also, in worms pre-treated with RNAi it is mentioned that 50 uM FuDR were added to PA14 plates, but wouldn't that affect PA14 proliferation and gut colonization? Could the authors elaborate on that?

> We appreciate the reviewer's comment. We performed slow-killing assays using mostly L4 stage animals on PA14-seeded plates with FUDR, following the methods used in many other previous papers (Reddy et al., 2009; Shiverse et al., 2010; Kirienko et al., 2014; Wu et al., 2018; Campos et al., 2021; Foster et al., 2020; Soo et al., 2021; Amrit et al., 2019; Jeong et al., 2020). In addition, for our fast-killing assays, we used PA14-seeded plates without FUDR, by following the methods used in various previous papers (Mahajan-Miklos et al., 1999; Tan et al., 1999; Adonizio et al., 2008; Kirienko et al., 2014; Soo et al., 2021; Cezairliyan et al., 2013; Jeong et al., 2020). Nevertheless, we agree with the reviewer that we need to clarify this issue for the readers. Therefore, we described the potential effect of FUDR on PA14 proliferation and gut colonization on the slow-killing assays that have been typically

used in the research field. We also mentioned that FUDR was not used for the fast-killing assays as follows.

Methods, page 20, line 412: “**PA14 slow-killing assays.** L4 (> 90%) and prefertile young adult (< 10%) stage animals that were pre-treated with control or *aco-2* RNAi bacteria on NGM plates or with fluoroacetic acid⁶⁹ (Sigma, St. Louis, MO, USA) were transferred to the PA14-seeded plates with 50 μM 5-fluoro-2-deoxyuridine (FUDR, Sigma, St. Louis, MO, USA), which prevents progeny from hatching; FUDR was used for standard slow killing assays in previous studies⁶⁸, but may affect proliferation and gut colonization of PA14.”

Methods, page 20, line 423: “**PA14 fast-killing assays.** PA14 fast-killing assays were performed as previously described with minor modifications^{15,68}. Briefly, 15 μl of PA14 liquid culture was spread on peptone-glucose-sorbitol (PGS) plates **without FUDR**, incubated for 24 hrs at 37°C, and kept at 25°C for eight hrs before use.”

4) In Extended Data figure 4C, what is the distinction between the two PMK-1 regulated clusters?

> We appreciate the reviewer’s comment. The previous Extended Data Fig. 4c (Supplementary Fig. 7a in this revised manuscript) displays two PMK-1-regulated clusters that were obtained by analyzing two different datasets from two papers (Fletcher et al., 2019; McEwan et al., 2016). We included the information in the Figure 2j and the Supplementary Figure 7 as follows.

Figure 2j legends, page 43, line 1006: “Genes upregulated in ^aWT vs. *pmk-1(km25)* [*pmk-1(-)*]⁸³, *vhp-1i* vs. Ctrlⁱ, *nipi-3(fr4)* [*nipi-3(-)*]⁸³ vs. WT, ^bWT vs.

*pmk-1(-)*⁸⁴, Ctrl*i* vs. *skn-1i*⁸⁵, WT vs. *atf-7(qd22 qd130)* [*atf-7(-)*]⁸⁴, *skn-1i*⁸⁵ vs. Ctrl*i*, *elt-2i*⁸⁶ vs. Ctrl*i*, and WT vs. *sek-1(km4)* [*sek-1(-)*]⁵⁵.”

Supplementary Figure 7, page 13, line 139: “^aWT vs. *pmk-1(-)*¹³, ^bWT vs. *pmk-1(-)*¹⁴. See Legends of Fig 2, Supplementary Fig 8, and Supplementary Table S3 for specific references of data sets used in these analyses.”

Also, in panel D, I can't find the circles representing the q-values.

> We removed circles representing q values because all data points had q values that are below 0.05. We included the information in the revised Supplementary Figure 7b as follows.

Supplementary Figure 7, page 13, line 137: “q values were obtained by calculating the false discovery rate corresponding to each normalized enrichment ($q < 0.05$ for all the comparisons).”

Also, if I understand correctly, here, as well as in Fig. 2, ELT-2 regulated genes examined are those that are UP-regulated after *elt-2* KD? i.e. ELT-2 repressed genes? and so are the NIP1-3 regulated genes.

I would not define that as indicating anything about immune signaling. Why not focus on genes that are down-regulated following KD of these immune regulators?

> The reviewer correctly pointed out that “*elt-2i* vs. Ctrl*i*” represented ELT-2-repressed genes and “*nipi-3(-)* vs. WT” represented NIP1-3-repressed genes in Figure 2j. Upon performing our comparison analysis, we found that Group i genes (798 genes upregulated by *aco-2* RNAi whose expression was increased by PA14 infection) substantially overlapped with genes that were downregulated by depletion of *elt-2* and *nipi-3*, but did not with genes that were upregulated by depletion of *elt-2*

or *nipi-3*, which are more likely immune signaling genes. Therefore, we focused on the functional characterization of other immune regulators in Figures 3 and 4, such as *pmk-1*, *atfs-1*, and *skn-1*, instead of *elt-2* and *nipi-3*. We clarified this issue in the Figure 2 legends as follows.

Figure 2 legends, page 43, line 1001: "Group i genes substantially overlapped with genes that were downregulated by depletion of *elt-2* and *nipi-3*, but did not with genes that were upregulated by depletion of *elt-2* or *nipi-3*, which are more likely immune signaling genes. We therefore focused on functionally characterizing other immune regulators in Figures 3 and 4, such as *pmk-1*, *atfs-1*, and *skn-1*, instead of *elt-2* and *nipi-3*."

Reviewer #2 (Remarks to the Author):

Kim et al. present a very interesting and timely manuscript that mitochondrial localized aconitase, by regulating the levels of oxaloacetate, can influence innate immunity. The manuscript is well written and the experiments are adequately defined and follow a logical progression that makes an excellent story. This manuscript is well-suited for Nature communications and will be of interest to the broad readership. In my opinion, the authors need only address one major concern and perhaps modify the text to address some minor issues. Overall, this is a very nice study that should be published following some suggested changes.

Major concerns:

the essential nature *aco-2* would suggest that even RNAi would make the animals sick and perhaps impact reproductive capacity. With this in mind the authors should

investigate the developmental impact and reproductive consequences of *aco-2* RNAi as these are likely to impact the immune response documented.

> We appreciate the reviewer's comment. We described the developmental and reproductive effects of *aco-2* RNAi in the Supplementary Figure 2 as follows.

Supplementary Figure 2, page 3, lines 31: "g. Representative images of control RNAi- and *aco-2* RNAi-treated animals 72 hrs after eggs were placed. Scale bar: 1 mm. *aco-2* RNAi reduced the total brood size of animals and slightly delayed development."

Developmental stage can greatly alter the response to pathogen and as such, animals with reduced ACO-2 and slower development might not be stage appropriately for the assays.

> We appreciate the reviewer's critical comment. Because we were well aware of the fact that developmental stage can alter the response to pathogen, we designed our experiments to minimize the effects of developmental differences on our results. First, we used only L4 animals for fast killing assays and mostly (over 90%) L4 animals for our slow killing assays to match the developmental stages of the animals. In addition, we assessed the developmental time of each RNA seq sample by using RAPToR [Real Age Prediction from Transcriptome staging on Reference (Bulteau and Francesconi, 2022)], and adjusted developmental time of the samples to that of the control samples for our RNA seq analysis. We now included these points in the Methods and the Supplementary Figure 6a as follows.

Methods, page 20, line 427: "**PA14 fast-killing assays.** ... L4 stage larval wild-type animals that were pre-treated with control or *aco-2* RNAi on NGM containing each of the Krebs cycle intermediates (8 mM, pH 6.0, based on a

previous study⁷) were transferred to PA14 on PGS plates for survival assays.”

Methods, page 20, line 412: “**PA14 slow-killing assays.** ... L4 (> 90%) and prefertile young adult (< 10%) stage animals that were pre-treated with control or *aco-2* RNAi bacteria on NGM plates or with fluoroacetic acid⁶⁹ (Sigma, St. Louis, MO, USA) were transferred to the PA14-seeded plates with 50 μ M 5-fluoro-2-deoxyuridine (FUDR, Sigma, St. Louis, MO, USA), which prevents progeny from hatching;”

Methods, on page 25, line 516: “**Analysis of RNA seq data.** ... Because mitochondrial dysfunctions generally slow development rates, which can be a confounding factor for the WormCat enrichment analysis, developmental time of each sample was estimated and adjusted to that of the control sample by using RAPToR [Real Age Prediction from Transcriptome staging on Reference⁷⁸].”

Supplementary Figure 5, page 10, line 103: “**a**, Developmental time of samples estimated from RNA seq data by using RAPToR [Real Age Prediction from Transcriptome staging on Reference⁹]. Developmental time of samples was adjusted to that of the control samples.”

Similarly, reduced reproductive output has been shown by Fred Ausubel's group to greatly impact innate immune responses.

> We appreciate the reviewer's comment. To minimize the possibility that reduced fertility by *aco-2* RNAi (Supplementary Figure 2g) contributed to increased immunity, we treated control RNAi- and *aco-2* RNAi-treated animals with *cdc-25.1* RNAi, which increases resistance against PA14 by inhibiting germline proliferation (Shapira and

Tan, 2008). We showed that *aco-2* RNAi enhanced survival on PA14 independently of *cdc-25.1* RNAi (Supplementary Figure 2f). Thus, reduced fertility caused by *aco-2* RNAi does not seem to be responsible for the enhanced resistance to PA14.

Results, page 5, line 89: “*aco-2* RNAi enhanced survival on PA14 independently of *cdc-25.1* RNAi (Supplementary Fig. 2f), which increases resistance against PA14 by reducing germline proliferation¹³. Thus, reduced fertility caused by *aco-2* RNAi (Supplementary Fig. 2g) does not seem to be responsible for the enhanced resistance to PA14.”

Supplementary Figure 2, page 3, line 29: “f, *aco-2* RNAi further increased the survival of *cdc-25.1* RNAi-treated worms that display sterility, which increases PA14 resistance¹, upon PA14 infection (small-lawn slow-killing assay).”

The authors should investigate how *aco-1* RNAi impacts *aco-2* and vice versa.

> We appreciate the reviewer’s comment. To address this issue, we performed qRT-PCR experiments using *aco-1* RNAi- and *aco-2* RNAi-treated animals. We found that *aco-1* RNAi decreased the level of *aco-1* mRNA levels, but did not affect the *aco-2* mRNA levels. We also showed that *aco-2* RNAi significantly reduced the *aco-2* mRNA levels, but did not alter the *aco-1* mRNA levels. Thus, *aco-1* RNAi and *aco-2* RNAi specifically knocked down the respective target genes, seemingly with no off-target or compensatory effects. We added these data to our revised manuscript by revising the Figure 1 legends, and the Supplementary Figure 2 as follows.

Figure 1 legends, page 40, line 945: “*aco-1* RNAi and *aco-2* RNAi respectively knocked down *aco-1* and *aco-2*, without off-target and compensatory effects (Supplementary Fig. 2e).”

Supplementary Figure 2, page 3, line 27: “e. *aco-2* RNAi reduced the mRNA level of *aco-2* while not altering that of *aco-1*, and *aco-1* RNAi decreased the mRNA level of *aco-1* but did not affect that of *aco-2* (n = 4).”

RNAi in neurons is very challenging, even in the enhanced neuronal RNAi strains.

The authors should quantify RNAi impact in this tissue perhaps with the GFP reporter).

> We appreciate the reviewer’s comment and agree with the reviewer. However, the reviewer #1 suggested removing the tissue-specific RNAi data from the manuscript (see above). Therefore, rather than quantifying RNAi impact in neurons, we removed the tissue-specific RNAi results from the manuscript.

Minor comments:

the authors should comment on the differential survival curves for WT and *aco-2* RNAi in figure 4. e.g., *aco-2* RNAi treated animals in Fig4d look like the WT in 4a and 4b for example.

> We appreciate the reviewer’s comment. As we previously described in Figure 4 legends and Supplementary Table S1, we performed small-lawn PA14 killing assays for Figure 4a-c, f, h, and I, and big-lawn PA14 killing assays for Figure 4d, e, and g. The main reason why we used two different methods for this Figure 4 is because during conducting this research project, the researchers who performed the PA14 survival assays were changed multiple times, because the researchers obtained their PhD degrees and moved on; for example, Yujin Lee and Dae-Eun Jeong performed experiments for Figure 4a-c, f, h, and I, whereas Hae-Eun Park, Sujeong Kwon and Yoonji Jung performed assays for Figure 4d, e, and g. Overall, we

performed the PA14 survival assays in the Figure 4 as a kind of small suppressor genetic screen for immune regulators. Thus, instead of repeating all the experiments again with the same platform, we described the differences in the small-lawn and big-lawn PA14 killing assays in Figure 4 legends and Supplementary Table S1. Following the reviewer's comment, we also described the difference in the survival curves for control RNAi- and *aco-2* RNAi-treated animals in the Figure 4 legends as follows.

Figure 4 legends, page 45, line 1027: "Small-lawn PA14 killing assays were performed for panels **a**, **b**, **c**, **f**, **h**, and **i**, whereas big-lawn PA14 killing assays were performed for panels **d**, **e**, and **g**; the survival curves for control RNAi- and *aco-2* RNAi-treated animals are different for these panels because of the difference in the methods: small-lawn vs. big-lawn PA14 killing assays."

The authors should take caution with epistatic relationship statements (same pathway, downstream, etc.). Most of the experiments in this manuscript are done with RNAi, which cannot be used for epistasis experiments.

> We appreciate the reviewer's comment. We carefully went through the text, and now do not mention the epistatic relationship statements with RNAi in the revised manuscript. Specifically, we toned down several sentences to be more cautious than before.

Result, page 8, line 145: "Next, we analyzed the immune signaling pathways that participated in the upregulation of pathogen responses by *aco-2* RNAi, by comparing Group i genes with previously published data (Fig. 2j; Supplementary Fig. 7,8)."

Result, page 9, line 166: “We investigated whether PMK-1/ATF-7 signaling or ATFS-1 **contributed to** the enhanced immunity caused by *aco-2* RNAi.”

Result, page 9, line 171: In addition, we found that **mutations in *atf-7*** **partially suppressed** the resistance of *aco-2* RNAi-treated animals against PA14 (Fig. 4d).”

Result, page 9, line 178: “These data suggest that ATFS-1, a key transcriptional regulator of the UPR^{mt} and immunity^{33–35} **contributes to** induced immune responses to PA14 by genetic inhibition of *aco-2*.”

References for this response letter

Bulteau, R., and Francesconi, M. Real age prediction from the transcriptome with RAPToR. 10.1101/2021.09.07.459270.

Cartwright-Acar, C.H., Osterhoudt, K., Suzuki, J.M.N.G.L., Gomez, D.R., Katzman, S., and Zahler, A.M. (2022). A forward genetic screen in *C. elegans* identifies conserved residues of spliceosomal proteins PRP8 and SNRNP200/BRR2 with a role in maintaining 5′ splice site identity. *Nucleic Acids Res.* 50, 11834–11857. 10.1093/nar/gkac991.

Chomyshen, S.C., Tabarraei, H., and Wu, C.W. (2022). Translational suppression via IFG-1/eIF4G inhibits stress-induced RNA alternative splicing in *Caenorhabditis elegans*. *Genetics* 221. 10.1093/genetics/iyac075.

Fletcher, M., Tillman, E.J., Butty, V.L., Levine, S.S., and Kim, D.H. (2019). Global transcriptional regulation of innate immunity by ATF-7 in *C. elegans*. *PLOS Genet.* 15, e1007830. 10.1371/journal.pgen.1007830.

McEwan, D.L., Feinbaum, R.L., Stroustrup, N., Haas, W., Conery, A.L., Anselmo, A., Sadreyev, R., and Ausubel, F.M. (2016). Tribbles ortholog NIPI-3 and bZIP transcription factor CEBP-1 regulate a *Caenorhabditis elegans* intestinal immune surveillance pathway. *BMC Biol.* 14, 105. 10.1186/s12915-016-0334-6.

Melo, J.A., and Ruvkun, G. (2012). Inactivation of Conserved *C. elegans* Genes Engages Pathogen- and Xenobiotic-Associated Defenses. *Cell* 149, 452–466. 10.1016/j.cell.2012.02.050.

Selak, M.A., Armour, S.M., MacKenzie, E.D., Boulahbel, H., Watson, D.G., Mansfield, K.D., Pan, Y., Simon, M.C., Thompson, C.B., and Gottlieb, E. (2005). Succinate links TCA cycle dysfunction to oncogenesis by inhibiting HIF- α prolyl hydroxylase. *Cancer Cell* 7, 77–85. 10.1016/j.ccr.2004.11.022.

Reddy, K.C., Andersen, E.C., Kruglyak, L., and Kim, D.H. (2009). A polymorphism in *npr-1* is a behavioral determinant of pathogen susceptibility in *C. elegans*. *Science*. 323, 382–384. 10.1126/science.1166527.

Shivers, R.P., Pagano, D.J., Kooistra, T., Richardson, C.E., Reddy, K.C., Whitney, J.K., Kamanzi, O., Matsumoto, K., Hisamoto, N., and Kim, D.H. (2010).

Phosphorylation of the conserved transcription factor ATF-7 by PMK-1 p38 MAPK regulates innate immunity in *Caenorhabditis elegans*. *PLoS Genet*. 6.10.1371/journal.pgen.1000892.

Kirienko, N. V., Cezairliyan, B.O., Ausubel, F.M., and Powell, J.R. (2014). *Pseudomonas aeruginosa* PA14 Pathogenesis in *Caenorhabditis elegans*. In *Methods in molecular biology* (Clifton, N.J.), pp. 653–669. 10.1007/978-1-4939-0473-0_50.

Campos, J.C., Wu, Z., Rudich, P.D., Soo, S.K., Mistry, M., Ferreira, J.C., Blackwell, T.K., and Van Raamsdonk, J.M. (2021). Mild mitochondrial impairment enhances innate immunity and longevity through ATFS-1 and p38 signaling. *EMBO Rep*. 22, 1–19. 10.15252/embr.202152964.

Foster, K.J., Cheesman, H.K., Liu, P., Peterson, N.D., Anderson, S.M., and Pukkila-Worley, R. (2020). Innate Immunity in the *C. elegans* Intestine Is Programmed by a Neuronal Regulator of AWC Olfactory Neuron Development. *Cell Rep*. 31, 107478. 10.1016/j.celrep.2020.03.042.

Soo, S.K., Traa, A., Rudich, P.D., Mistry, M., and van Raamsdonk, J.M. (2021). Activation of mitochondrial unfolded protein response protects against multiple exogenous stressors. *Life Sci. Alliance* 4, 1–16. 10.26508/lsa.202101182.

Amrit, F.R.G., Naim, N., Ratnappan, R., Loose, J., Mason, C., Steenberge, L., McClendon, B.T., Wang, G., Driscoll, M., Yanowitz, J.L., et al. (2019). The longevity-promoting factor, TCER-1, widely represses stress resistance and innate immunity. *Nat. Commun*. 10, 3042. 10.1038/s41467-019-10759-z.

Jeong, D.-E., Lee, Y., Ham, S., Lee, D., Kwon, S., Park, H.-E.H., Hwang, S.-Y., Yoo, J.-Y., Roh, T.-Y., and Lee, S.-J. V (2020). Inhibition of the oligosaccharyl transferase in *Caenorhabditis elegans* that compromises ER proteostasis suppresses p38-dependent protection against pathogenic bacteria. *PLOS Genet*. 16, e1008617. 10.1371/journal.pgen.1008617.

Mahajan-Miklos, S., Tan, M.W., Rahme, L.G., and Ausubel, F.M. (1999). Molecular mechanisms of bacterial virulence elucidated using a *Pseudomonas aeruginosa*-*Caenorhabditis elegans* pathogenesis model. *Cell* 96, 47–56. 10.1016/S0092-8674(00)80958-7.

Tan, M.W., and Ausubel, F.M. (2000). *Caenorhabditis elegans*: A model genetic host to study *Pseudomonas aeruginosa* pathogenesis. *Curr. Opin. Microbiol*. 3, 29–34. 10.1016/S1369-5274(99)00047-8.

Adonizio, A., Leal, S.M., Ausubel, F.M., and Mathee, K. (2008). Attenuation of *Pseudomonas aeruginosa* virulence by medicinal plants in a *Caenorhabditis elegans* model system. *J. Med. Microbiol.* 57, 809–813. 10.1099/jmm.0.47802-0.

Cezairliyan, B., Vinayavekhin, N., Grenfell-Lee, D., Yuen, G.J., Saghatelian, A., and Ausubel, F.M. (2013). Identification of *Pseudomonas aeruginosa* Phenazines that Kill *Caenorhabditis elegans*. *PLoS Pathog.* 9. 10.1371/journal.ppat.1003101.

Shapira, M., and Tan, M.-W. (2008). Genetic Analysis of *Caenorhabditis elegans* Innate Immunity. In *Innate Immunity* (Humana Press), pp. 429–442. 10.1007/978-1-59745-570-1_25.

REVIEWER COMMENTS

Reviewer #1 (Remarks to the Author):

The authors addressed most of my comments well, except for one minor point, regarding the potential effects of FuDR on pathogen proliferation and colonization.

The authors point out that FuDR was added only in slow killing assays, but not in fast-killing assays. This does not resolve the issue, since FuDR should not affect fast killing assays, where phenotype is independent of pathogen colonization (only toxin secretion). The issue is in slow killing assays. Overall, the time course of killing seems to indicate that pathogen colonization proceeds as intended; nevertheless, I think it would be better if the authors demonstrated that *aco-2*(RNAi) has the same effects without FuDR.

Reviewer #2 (Remarks to the Author):

The authors have addressed my major concerns.

REVIEWER COMMENTS

Reviewer #1 (Remarks to the Author):

The authors addressed most of my comments well, except for one minor point, regarding the potential effects of FuDR on pathogen proliferation and colonization. The authors point out that FuDR was added only in slow killing assays, but not in fast-killing assays. This does not resolve the issue, since FuDR should not affect fast killing assays, where phenotype is independent of pathogen colonization (only toxin secretion). The issue is in slow killing assays. Overall, the time course of killing seems to indicate that pathogen colonization proceeds as intended; nevertheless, I think it would be better if the authors demonstrated that *aco-2*(RNAi) has the same effects without FuDR.

> We appreciate the reviewer's comment. Following the suggestion by the reviewer, we performed PA14 slow-killing assays without FUDR, using control RNAi- and *aco-2* RNAi-treated animals. In two independent experiments, we found that *aco-2* RNAi substantially increased resistance against PA14 without FUDR (177% and 206%: average 192%). Thus, *aco-2* RNAi increases the resistance against PA14 infection independently of FUDR. We included these new data in our revised manuscript by revising the Figure 1 as follows.

Result, page 5, line 95: "In addition, *aco-2* RNAi increased the survival of animals on the big lawn of PA14 (Fig. 1h,i), where the animals cannot avoid the pathogen¹⁴, and under a fast-killing condition (Fig. 1j), elicited by toxins secreted from PA14¹⁵."

Figure 1 legends, page 40, line 958: "**h–k**, *aco-2* RNAi increased the survival of animals under big-lawn PA14-mediated slow-killing with (**h**) or without (**i**) 5-fluoro-

2'-deoxyuridine (FUDR) treatment, and fast-killing (j) and *S. aureus* infection (k) conditions.”

REVIEWERS' COMMENTS

Reviewer #1 (Remarks to the Author):

The authors have addressed all my comments. The manuscript looks great, and as far as I'm concerned, it's ready for publication.